# MoDr: Mixture-of-Depth-Recurrent Transformers for Test-Time Reasoning

**Xiaojing Zhang[1], Haifeng Wu[3], Gang He[1], Jiyang Shen[1], Bochen Lyu[†,2,1], Zhanxing Zhu[†,2]**
[1]DataCanvas    [2]University of Southampton    [3]Find AI

## Abstract

Large Language Models have demonstrated superior reasoning capabilities by generating step-by-step reasoning in natural language before deriving the final answer. Recently, Geiping et al. (2025) introduced 3.5B-Huginn as an alternative to this paradigm, a depth-recurrent Transformer that increases computational depth per token by reusing a recurrent block in latent space. Despite its performance gains with increasing recurrences, this approach is inadequate for tasks demanding exploration and adaptivity, a limitation arising from its single, chain-like propagation mechanism. To address this, we propose a novel dynamic multi-branches routing approach for Huginn, termed as **M**ixture-**of**-**D**epth-**R**ecurrent (MoDr) Transformer, which enables effective exploration of the solution space by shifting linear latent reasoning into a LoRA-based multi-branch dynamic relay mode with a learnable hard-gate routing. Meanwhile, we introduce an auxiliary-loss-free load balancing strategy to mitigate the potential routing collapse. Our empirical results reveal that MoDr achieves average accuracy improvements of +7.2% and +2.48% over the original Huginn model and its fine-tuned variant, respectively, across various mathematical reasoning benchmarks and improvements of +21.21% and +1.52% on commonsense reasoning benchmarks.

## 1 Introduction

Transformer-based large language models (LLMs) (Achiam et al., 2023; Team et al., 2023; Liu et al., 2024) have achieved striking performance gains in a wide variety of reasoning tasks (Gao et al., 2023; Srivastava et al., 2023; Rein et al., 2024). To enhance the reasoning capabilities of LLMs, many studies have focused on scaling both model size (Wei et al., 2022a) and test-time computation (Ji et al., 2025). However, much of this success comes at the cost of significant computational and memory resources during training and deployment, and is accompanied by high latency as the model verbalizes excessively long intermediate reasoning (referred to as "deep thinking") before delivering a final response.

Recent developments in latent reasoning, which leverage recurrent or looped-based methods, have shown potential to improve inference efficiency apart from scaling model size and employing explicit thinking (Dehghani et al., 2019; Gao et al., 2024b; Chen et al., 2025; Geiping et al., 2025). Among these, Geiping et al. (2025) proposed a 3.5B depth-recurrent transformer, dubbed Huginn, which explicitly disentangles the transformer into a three-stage Prelude/Loop/Coda structure (as shown in Figure 1(a)), in order to reuse the rumination module (Loop) multiple times in the latent space to increase the computational depth per token. While the recurrent layers achieve low memory and latency alongside excellent reasoning performance, its single recurrent reasoning module limits the adaptability of the reasoning trajectory.

From the perspective of thought structure, the chain structure (Wei et al., 2022b) significantly enhances the reasoning capabilities of LLMs compared to directly providing the answer. Subsequently, the tree structure (Yao et al., 2023) further strengthens the model's ability to explore and backtrack by introducing hierarchical search. Furthermore, the graph structure (Besta et al., 2024), by incorporating loops and N-to-1 connections, facilitates sub-problem aggregation and self-verification.

---

[†] Correspondence to: {`bochen.lyu, z.zhu`}`@soton.ac.uk`.
Code is available at: https://github.com/zhangxjohn/MoDr.

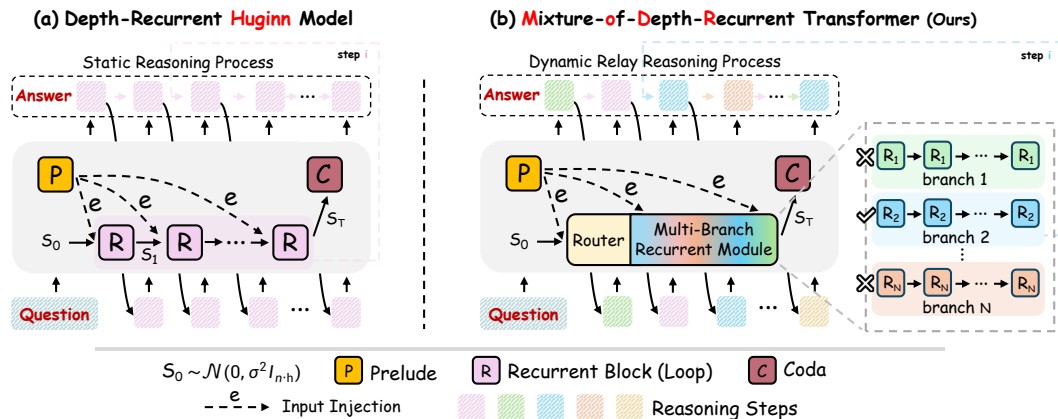

Figure 1: Comparison of (a) vanilla Huginn (Geiping et al., 2025) model and (b) MoDr (ours) reasoning patterns. Unlike Huginn, where each reasoning step relies on a single recurrent module, our proposed MoDr employs a dynamic routing mechanism to adaptively select the most suitable recurrent branch for the current context input, thereby predicting the next token. During inference, in Huginn, each token is generated by a fixed recurrent module, whereas in MoDr, tokens are produced dynamically through a sequential relay of multiple recurrent branches.

Inspired by these works, we hypothesize that a single recurrent reasoning module analogous to the chain structure inherently limits the scope of exploration. This inspires us to explore the question: How can we construct a depth-recurrent Huginn model with an adaptive exploration-rumination module while avoiding extra resource burdens?

To this end, we propose an innovative approach, **M**ixture-**o**f-**D**epth-**R**ecurrent (MoDr) Transformer, which conceptualizes the reasoning process as a dynamic relay exploration for each token in a combinatorial solution space (see Figure 1(b) and 2 for illustration). Specifically, we first introduce several block-wise low-rank adapters (LoRAs (Hu et al., 2022)) as versatile exploration branches. Each branch is integrated with the shared rumination recurrent module (Loop). Considering that different contextual inputs may have diverse computational demands for reasoning path exploration, a hard-gate routing mechanism is then employed to comprehensively consider the hidden state information of the context, and automatically decide which branch is for predicting the next token. In addition, we utilize an auxiliary-loss-free load balancing strategy to mitigate the risk of routing collapse (Shazeer et al., 2017), inspired by Wang et al. (2024). Empirically, our comprehensive experiments across a wide range of mathematical and commonsense reasoning benchmarks validate the effectiveness of the MoDr approach.

**Contributions.**

- We find that the inherent limitation of existing depth-recurrent Huginn model lies in its rumination module (the 'Loop'), which adopts a single, chain-like propagation mechanism. This design weakens the diversity and exploration capability of the model's reasoning trajectories within the latent space.

- We introduce **M**ixture-**o**f-**D**epth-**R**ecurrent (MoDr) Transformer, a novel extension of the depth-recurrent Huginn architecture designed to shift latent reasoning into a multi-branch dynamic relay exploration mode with negligible resource overhead.

- We conduct extensive experiments on a wide variety of mathematical and commonsense reasoning benchmarks, demonstrating that MoDr achieves competitive performance. Specifically, compared to the vanilla Huginn model and its fine-tuned variant, MoDr achieves average accuracy improvements of +7.2% and +2.48% on mathematical tasks, and +21.21% and +1.52% on commonsense tasks, respectively.

## 2 BACKGROUND

To scale test-time computation while reducing both training and deployment computational overhead, a prevailing line of research is to enable depth adaptivity in Transformers through recurrent depth and looped layers. Starting from the Universal Transformer (Dehghani et al., 2019), which pioneered dynamic recurrence over layers to iteratively refine sequence representations, this design paradigm has demonstrated that depth-adaptive reasoning is a promising substitute for the traditional fixed-depth transformer architecture. Subsequently, research on the Looped Transformers has exhibited strong generalization performance across diverse tasks, including programmable computing (Giannou et al., 2023), data fitting (Yang et al., 2023), and arithmetic reasoning (Saunshi et al., 2025). In addition to adopting monolithic recurrent designs, AlgoFormer (Gao et al., 2024b) and Depth-Recurrent Huginn (Geiping et al., 2025) proposed a three-stage Prelude/Loop/Coda structure, which can be formulated as:

$$f = f_{\text{head}} \circ f_{\text{coda}} \circ \underbrace{f_R \circ \cdots \circ f_r \circ \cdots \circ f_1}_{T \text{ iterations}} \circ f_{\text{prelude}} \circ f_{\text{embed}}, \tag{1}$$

with $R$ hidden layers involved in the Loop for $T$ times. Note that $f_{\text{pre}}$ and $f_{\text{coda}}$ could have more than one hidden layer. The modularization of this architecture can provide efficiency and adaptability while generalizing across a broader range of applications. In addition, depth-recurrent transformers increase computational depth per token by reusing intermediate layers, which can be viewed as "deep thinking" in the continuous latent space to facilitate reasoning.

### 2.1 PRELIMINARY: DEPTH-RECURRENT HUGINN

We first outline the architecture of the model that our framework is built upon, termed Huginn, which is a 3.5B scalable recurrent decoder-only transformer (Figure 1(a)). The model is primarily structured around three functional modules: (1) 2 *prelude* blocks, which are responsible for embedding input context into a latent space; (2) 4 *recurrent* blocks, which sequentially process the output from the *prelude* module; (3) 2 *coda* blocks, which decode from the latent space to predict the next token. Notably, all blocks follow standard transformer layer design, and each layer contains a multi-head causal self-attention mechanism.

Concretely, given a sequence of tokens $\mathbf{x} = [x_1, x_2, ..., x_n]$, where $x_i \in \mathbb{R}^{|V|}$, $n$ denotes the length of the input context, and $|V|$ represents the size of the vocabulary. Unlike the forward pass of a standard Transformer, the model explicitly separates its computation flow into input encoding $\mathbf{P}$ (*prelude* module), iterative implicit reasoning $\mathbf{R}$ (*recurrent* module), and output decoding $\mathbf{C}$ (*coda* module). These modules ultimately produce output probabilities $\mathbf{p} \in \mathbb{R}^{n \times |V|}$ as follows:

$$\mathbf{e} = \mathbf{P}(\mathbf{x}), \tag{2}$$
$$\mathbf{s}_0 \sim \mathcal{N}(0, \sigma^2 \mathbf{I}_{n \cdot h}), \tag{3}$$
$$\mathbf{s}_t = \mathbf{R}(\mathbf{e}, \mathbf{s}_{t-1}) \quad \text{for} \quad t \in \{1, 2, ..., T\}, \tag{4}$$
$$\mathbf{p} = \mathbf{C}(\mathbf{s}_T), \tag{5}$$

where $\mathbf{s}_0$ is a random Gaussian vector serving as the initial state of the recurrent module, $\sigma$ is some standard deviation, and $h$ is the hidden dimension. During the $T$ recurrent steps, the model repeatedly applies the core unit $\mathbf{R}$, which takes the latent state $\mathbf{s}_{t-1}$ and the embedded input $\mathbf{e}$ as input, and outputs the updated latent state $\mathbf{s}_t$. Although this model only includes 8 trainable blocks (i.e., $2\mathbf{P} + 4\mathbf{R} + 2\mathbf{C}$), it allows for an infinitely deep transformer in the limit as $T \to \infty$. Note that while the depth-recurrent Huginn gains efficiency by using recurrent states in the latent space instead of generating explicit reasoning steps, its reliance on a single recurrent module for all reasoning steps fundamentally constrains its reasoning capabilities and flexibility.

### 2.2 DYNAMIC ROUTING AND PARAMETER-EFFICIENT FINE-TUNING

Recent studies have established Mixture-of-Experts (MoE) as a prominent architecture for scaling LLMs (Liu et al., 2024; Yang et al., 2025). Its key feature is that MoE models can dynamically select the most suitable expert from multiple candidates for different inputs. This approach achieves a favorable trade-off between model capability and computational efficiency while maintaining a

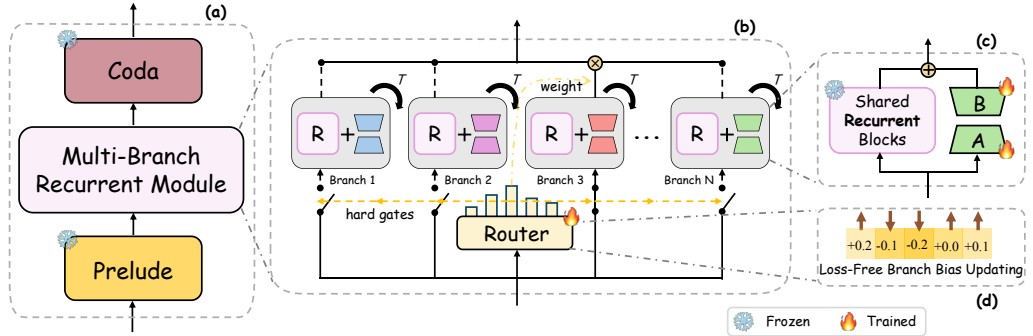

Figure 2: The architecture of the **M**ixture-**o**f-**D**epth-**R**ecurrent (MoDr) Transformer. MoDr consists of N recurrent branches, which are formed by combining the original recurrent blocks of Huginn with different LoRAs, with the weights of the original recurrent blocks shared across all branches. In addition, a hard-gate routing mechanism dynamically selects the appropriate branch for the current input information. During fine-tuning, only the LoRA branches and the router are trained, and an auxiliary-loss-free balancing mechanism ensures even load distribution across different branches.

nearly constant number of FLOPs per token for each expert (Zoph et al., 2022). Instead of applying MoE directly during pre-training, methods such as MoELORA (Luo et al., 2024), MOLA(Gao et al., 2024a), and MIXLORA (Li et al., 2024) enhance multi-task generalization by integrating multiple LoRA adapters as *layer-wise* experts within transformer sub-layers during fine-tuning. Furthermore, LoTA (Panda et al., 2024) mitigates destructive interference by optimizing a sparse sub-network of the model. As noted earlier, the inflexibility of Huginn in generating each token stems from its rigid recurrent module. To address this flaw, we propose MoDr, a novel MoE+LoRA architecture that departs from the conventional layer-wise paradigm. Particularly, MoDr integrates *block-wise* low-rank adapters directly into Huginn's recurrent module and couples them with a dynamic routing mechanism. This design fundamentally enhances the adaptability and richness of the model's reasoning trajectories, establishing a new direction for adaptive reasoning in recurrent-depth models. We discuss more related works in Appendix A.1.

## 3 METHODOLOGY

To enable adaptive latent space reasoning, in this work, we propose an innovative approach, **M**ixture-**o**f-**D**epth-**R**ecurrent (MoDr) Transformer, which fine-tunes the existing Huginn model to establish a multi-branch, dynamic reasoning pathway. This section outlines the architecture and methodology of MoDr, as illustrated in Figure 2. Specifically, we first integrate multiple low-rank adapters (Lo-RAs) to construct distinct recurrent reasoning branches, all of which share the weights of the original recurrent blocks in the Huginn model (§3.1). Then, we present a hard-gate routing mechanism that takes into account the hidden state information of the current context to select which branch will predict the next token (§3.2). To alleviate the load imbalance among different branches, we adopt an auxiliary-loss-free load balancing strategy to mitigate uneven load distribution across branches during training (§3.3).

### 3.1 LORA-BASED MULTI-BRANCH RECURRENT MODULE

As discussed above, Huginn broadly follows the design of standard transformer layers. Accordingly, the architecture of Huginn's core recurrent blocks is built upon the standard transformer "sandwich" structure, which sequentially stacks a multi-head causal attention layer (Attn) and a multilayer perceptron (MLP), with each sub-layer featuring a residual connection and layer normalization (LN). Let $\mathbf{z}^l \in \mathbb{R}^{n \times h}$ denote the hidden state output by the $l$-th recurrent block. For each recurrent step $t \in \{1, 2, ..., T\}$, the hidden state $\mathbf{z}_t^l$ is computed as:

$$\hat{\mathbf{z}}_t^l = \text{LN}(\text{Attn}(\text{LN}(\mathbf{z}_t^{l-1})|\mathbf{W}^l) + \mathbf{z}_t^{l-1}), \tag{6}$$

$$\mathbf{z}_t^l = \text{LN}(\text{MLP}(\text{LN}(\hat{\mathbf{z}}_t^l)|\mathbf{W}^l) + \hat{\mathbf{z}}_t^l), \tag{7}$$

where $\mathbf{W}^l$ denotes the parameters of the $l$-th recurrent block. Our objective is to construct an adaptive exploration-rumination module. Here, a key challenge arises: how can we break free from the rigidly sequential forward propagation constraints inherent in existing recurrent architecture?

To address this, we propose to utilize different branches to dynamically take turns to predict the next token according to the current context information, until the end of reasoning. Full fine-tuning with multiple initialized instances of a recurrent module in Huginn can enhance diversity and exploration within the model's latent reasoning trajectories. However, this approach incurs substantial computational and memory overhead. To mitigate these costs, MoDr integrates multiple low-rank adapters (LoRAs) to create distinct recurrent reasoning branches, with the weights of the original recurrent blocks being shared across all branches, as shown in Figure 2(c). During training, only the LoRA weights of all branches are fine-tuned. This design yields two primary advantages: (1) the backbone model is frozen to preserve its world knowledge, and (2) LoRA introduces negligible computational or memory overhead owing to its minimal number of trainable parameters. Therefore, the output hidden state $\mathbf{z}_t^l$ of each recurrent branch is computed as follows:

$$\hat{\mathbf{z}}_{j,t}^l = \mathrm{LN}(\mathrm{Attn}(\mathrm{LN}(\mathbf{z}_{j,t}^{l-1})|\mathbf{W}^l, \Delta\mathbf{W}_j^l) + \mathbf{z}_{j,t}^{l-1}), \tag{8}$$

$$\mathbf{z}_{j,t}^l = \mathrm{LN}(\mathrm{MLP}(\mathrm{LN}(\hat{\mathbf{z}}_{j,t}^l)|\mathbf{W}^l, \Delta\mathbf{W}_j^l) + \hat{\mathbf{z}}_{j,t}^l), \tag{9}$$

where $\{\Delta\mathbf{W}_j^l\}_{j=1}^N$ denotes the trained LoRA module parameters for $N$ recurrent branches. Specifically, for a base feature transformation $\mathbf{z} = \mathbf{W}_0\mathbf{x}$, our modified forward pass yields:

$$\mathbf{z} = \mathbf{W}_0\mathbf{x} + \frac{\alpha}{r}\Delta\mathbf{W}\mathbf{x} = \mathbf{W}_0\mathbf{x} + \frac{\alpha}{r}\mathbf{B}\mathbf{A}\mathbf{x}, \tag{10}$$

where $\mathbf{B} \in \mathbb{R}^{h \times r}$ and $\mathbf{A} \in \mathbb{R}^{r \times k}$ with rank $r \ll \min(h, k)$ for $h$ and $k$ being the dimensions of the original parameter matrix $\mathbf{W}_0$. The scaling factor $\alpha$ controls the adaptation magnitude.

## 3.2 HARD-GATE BRANCH ROUTING STRATEGY

Inspired by sparely-gated Mixture-of-Experts (MoE) (Shazeer et al., 2017) and switch Transformer (Fedus et al., 2022), we design a learnable hard-gate routing network to determine which candidate recurrent branch will predict the next token according to the hidden state information. In our method, the hidden state information $\mathbf{h} \in \mathbb{R}^{n \times h}$ derives from two aspects: (1) the output $\mathbf{e}$ of the prelude blocks, and (2) the recurrent state $\mathbf{s}$. We utilize an adapter matrix: $\mathbb{R}^{2h} \to \mathbb{R}^h$ to map the concatenation of $\mathbf{e}$ and $\mathbf{s}$ into the hidden dimension $h$.

Let $\mathbf{W}_{\mathrm{router}} \in \mathbb{R}^{N \times h}$ denote the trainable weight matrix of a routing network, where $N$ is the number of candidate recurrent branches. As depicted in Figure 2(b), our design employs a Top-K hard-gate router. For the sake of clarity, we use Top-1 as an example to illustrate this. This router selects the single branch that achieves the highest average confidence score across the tokens, as determined by the current context information. The process is detailed below:

$$\mathbf{u} = \mathbf{W}_{\mathrm{router}}\mathbf{h}^\top, \quad \mathbf{u} \in \mathbb{R}^{N \times n}, \tag{11}$$

$$\boldsymbol{r} = \sigma(\frac{1}{n}\sum_{i=1}^n (\boldsymbol{u}_i)), \quad \boldsymbol{r} \in \mathbb{R}^N, \tag{12}$$

$$\zeta = \mathrm{argmax}_j(r_j), \quad j \in \{1, 2, ..., N\}, \tag{13}$$

$$g = r_j, \quad \mathrm{if} \quad j = \zeta, \tag{14}$$

where $\zeta$ is the index of the selected recurrent branch, $g$ is a scalar score, and $\sigma$ is a nonlinear activation function like sigmoid or softmax. After that, all current hidden states $\mathbf{z}_{j,t}^l$ of the selected branch are weighted as $\mathbf{z}_{j,t}^{l'} = g \cdot \mathbf{z}_{j,t}^l$.

During inference, for each newly generated token, the hard-gate router dynamically selects which recurrent branch will perform the "deep thinking" based on the contextual information from the previous reasoning steps. This next-token prediction process resembles a "relay race" across different branches. For an illustration, see Figure 1(b).

### 3.3 Auxiliary-Loss-Free Load Balance

To prevent imbalanced training among branches caused by routing collapse (Shazeer et al., 2017), we introduce a load balancing strategy from a sequence-wise perspective. Existing studies typically employ an auxiliary loss (Lepikhin et al., 2020; Fedus et al., 2022) to address load imbalance in the MoE system during training. However, evidence suggests that a large auxiliary loss could introduce significant conflicting gradients into training, thereby degrading model performance (Wang et al., 2024). To overcome this challenge, we introduce a loss-free balancing approach to directly adjusts the gate scores of candidate branches based on their individual load conditions.

As illustrated in Figure 2(d), we add a bias term $\{b_i\}_{i=1}^N$ to the original gating score $\{r_i\}_{i=1}^N$ of each candidate branch. The branch selection and the final weight score are then computed as follows:

$$\hat{\boldsymbol{r}} = \boldsymbol{r} + \boldsymbol{b}, \quad \hat{\boldsymbol{r}} \in \mathbb{R}^N, \tag{15}$$

$$\hat{\zeta} = \operatorname{argmax}_j(\hat{r}_j), \quad j \in \{1, 2, ..., N\}, \tag{16}$$

$$g = r_j, \quad \text{if} \quad j = \hat{\zeta}, \tag{17}$$

where $\hat{\zeta}$ denotes the index of the selected recurrent branch after the bias term adjustment routing strategy. Note that the corresponding weight score $g$ does not involve the bias term $b_j$.

To adjust the per-branch bias $b_i$ ($i \in \{1, 2, ..., N\}$) during training, each bias $b_i$ is first initialized to 0. For each batch, the number of assigned samples $c_i$ per branch and their average number $\bar{c}_i$ are counted. Then, the load violation error $e_i$ is computed, and the bias $b_i$ is updated as follows:

$$e_i = \bar{c}_i - c_i, \tag{18}$$

$$b_i = b_i + \eta * \operatorname{sign}(e_i), \tag{19}$$

where $\eta$ is the update rate of the bias term, and $\operatorname{sign}(\cdot)$ is a sign function. This load balancing strategy not only enables load balancing across branches but also prevents the direct incorporation of noisy gradients into the model.

## 4 Experiments

In this section, we evaluate the effectiveness of our proposed MoDr, with a focus on mathematical reasoning tasks. We present additional experiments for commonsense reasoning tasks in Appendix A.2. All experiments are conducted on a single NVIDIA Tesla H100 GPU with 80GB of VRAM.

### 4.1 Experimental Setup

**Datasets.** We conduct our experiments on six mathematical reasoning tasks: (1) **GSM8K** (Cobbe et al., 2021), a dataset of high quality grade school math word problems. (2) **MAWPS** (Koncel-Kedziorski et al., 2016) dataset, a curated online repository of arithmetic and algebra word problems. (3) **AQuA** (Ling et al., 2017) dataset, focusing on algebraic word problems. (4) **MultiArith** (Roy & Roth, 2016), containing multi-step mathematical word problems. (5) **AddSub** (Hosseini et al., 2014), a dataset of addition and subtraction arithmetic word problems. (6) **SingleEq** (Koncel-Kedziorski et al., 2015), covering grade-school algebra word problems. Regarding the above datasets, only GSM8K, MAWPS, and AQuA provide training sets, whereas MultiArith, AddSub, and SingleEq serve as three out-of-domain benchmarks to evaluate the models' robustness. To enhance the reasoning capabilities of the fine-tuned models, particularly their step-by-step rationales, we employed Qwen2.5-Math-7B-Instruct [1] to generate chain-of-thought reasoning steps

---

[1] https://huggingface.co/Qwen/Qwen2.5-Math-7B-Instruct

for the training sets. To ensure data quality, we removed samples with incorrect answers. Table 1 summarizes the detailed statistics of the datasets. The accuracy is the evaluation metric used across all benchmarks to measure the correctness of the predicted answers.

Table 1: Statistics of Mathematical Reasoning Datasets.

| Dataset | In Domain | | | Out of Domain | | |
|---|---|---|---|---|---|---|
| | GSM8K | MAWPS | AQuA | MultiArith | AddSub | SingleEq |
| Answer Type | Number | Number | Option | Number | Number | Number |
| # Train Sample | 7130 | 1826 | 609 | - | - | - |
| # Test Sample | 1319 | 238 | 254 | 600 | 395 | 508 |

**Baselines.** We primarily compare against the following baselines: First, the vanilla Huginn model (Geiping et al., 2025), which constitutes our base architecture for MoDr. Second, we introduce a LoRA-based supervised fine-tuning (SFT) variant of this base model, termed Huginn-SFT, using hyperparameters identical to those of MoDr. Furthermore, we evaluate a multi-branch Huginn model without a router (i.e., using random branch selection) to assess the router's contribution to performance.

**Settings.** For all experiments in this section, we use the AdamW optimizer with a learning rate of 4e-5, weight decay of 1e-4, and betas set to (0.9, 0.95). We clip gradients with a threshold of 0.2 and employ a cosine learning rate schedule with 10% warmup. Models are trained using a batch size of 4 and a sequence length of 512, and the number of epochs matches the number of recurrent branches. To scale the depth-recurrent architecture, we train with a mean recurrence value of 32. To reduce computational and memory costs during training, we truncate backpropagation through time (BPTT) to the last 8 iterations of the recurrent unit. For all low-rank adapters associated with each branch, both the rank and the scaling factor are configured to be 16. These adapters activate the $q, k, v, o$ projections in attention layers of the *recurrent* blocks. Regarding the branch routing and load balancing strategies, we use a sigmoid nonlinear activation function and set the bias term's update rate to 0.001. Among the routed branches, a single branch is activated per token.

## 4.2 MAIN RESULTS

### 4.2.1 MATHEMATICAL REASONING

In this experiment, MoDr is equipped with four LoRA-based recurrent branches and a hard-gate router. Its trainable parameters constitute less than 0.2% of the base Huginn model. Our baseline, Huginn-SFT, is a conventional single-branch model fine-tuned on the same datasets and settings. Figure 3 presents a comparison between the baseline models and our proposed MoDr. The results demonstrate that this multi-branch architecture significantly boosts reasoning capabilities with a negligible parameter overhead. Notably, across six mathematical reasoning datasets, MoDr achieves average accuracy improvements of +7.2% and +2.48% over the vanilla Huginn model and its fine-tuned variant, respectively. This advantage stems from the fact that MoDr can dynamically route the hidden state from the prelude module to the most suitable branch for next-token prediction, thereby enhancing the adaptability of the reasoning trajectory.

To assess robustness, we split the six benchmarks into three in-domain (ID) (i.e., GSM8K, MAWPS, and AQuA) and three out-of-domain (OOD) (i.e., MultiArith, AddSub, and SingleEq) tasks. We can also observe that MoDr exhibits a superior performance relative to all baseline models, irrespective of the domain setting. Specifically, MoDr excels even more on out-of-domain datasets (i.e., outperforming the original Huginn and Huginn-SFT by +6.75% and +1.78% on the ID, and by +7.67% and +3.18% on the OOD, respectively), suggesting strong generalization. Overall, these results well demonstrate the effectiveness of our proposed MoDr.

### 4.2.2 COMMONSENSE REASONING

We also evaluated MoDr on various commonsense reasoning benchmarks. As reported in Figure 7 of Appendix A.2, our findings reveal that MoDr achieves the most significant performance gains over both the vanilla Huginn and the Huginn-SFT models. This indicates the broad applicability of MoDr for boosting LLM's performance. See Appendix A.2 for more details.

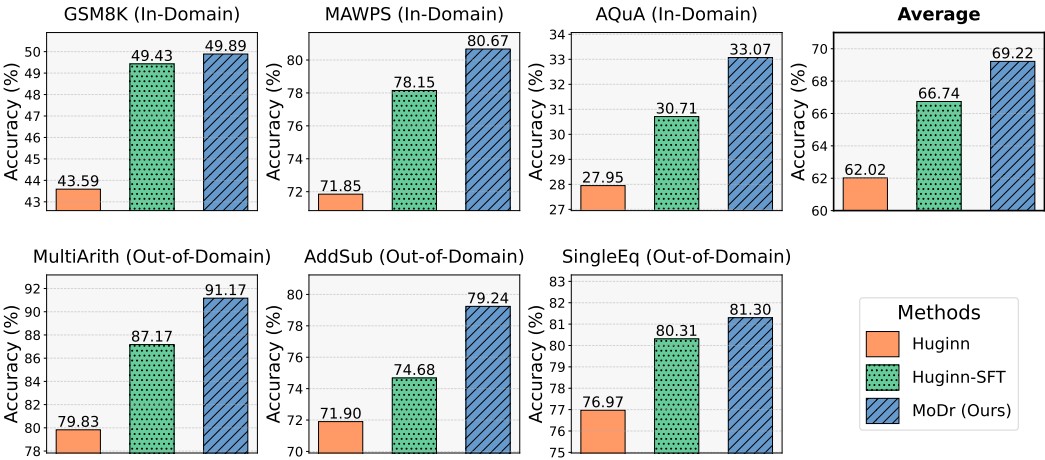

Figure 3: Performance comparison of MoDr with baseline methods on in-domain and out-of-domain mathematical reasoning benchmarks. The "Average" represents the overall mean of the six mathematical datasets.

## 4.3 ABLATION STUDY

**Impact of Router.** We investigated the impact of dynamic routing from two perspectives. On one hand, during inference, we deliberately disabled the router in MoDr and employed random branch selection (referred to as MoDr w/o Router). On the other hand, we directly fine-tuned a model without a router, the 4-branch Huginn, which also uses random branch selection in both the training and inference stages (referred to as No Router). As shown in Table 2, MoDr with dynamic routing demonstrates superior inference performance over both router-free models across all mathematical benchmarks.

In addition, two observations are worth noting: (1) No Router performs best on GSM8K, but only marginally. This can be attributed to the fact that GSM8K's extensive training data (74.52%) thoroughly trains the model branches in all setups. (2) The performance of No Router is better than that of MoDr w/o Router but is still inferior to that of the standard MoDr with its router. This further underscores the positive role of dynamic routing in branch selection during inference.

Table 2: Performance comparison among three configurations: a 4-branch Huginn model without a router (i.e., using random branch selection), a MoDr model with a router, and a MoDr model with a disabled router (i.e., it also uses a random strategy to select a branch per token). The top score in each column is in **bold**.

| Method | **In Domain** | | | **Out of Domain** | | | Average |
|---|---|---|---|---|---|---|---|
| | GSM8K | MAWPS | AQuA | MultiArith | AddSub | SingleEq | |
| No Router (random) | **50.72** | 79.41 | 31.89 | 90.17 | 75.70 | 76.57 | 67.41 |
| MoDr w/ Router (Ours) | 49.89 | **80.67** | **33.07** | **91.17** | **79.24** | **81.30** | **69.22** |
| ↳ w/o Router (random) | 48.60 | 77.73 | 29.92 | 89.17 | 74.68 | 78.35 | 66.41 |

**Impact of Single Branch.** To validate the effectiveness of dynamic routing for multi-branch inference, we conducted an ablation study by evaluating each individual branch of the 4-branch MoDr independently on all mathematical tasks. The results, presented in Table 3, demonstrate that the average accuracy of MoDr with dynamic routing is superior to that of any individual branch and even exceeds their collective average. Notably, MoDr with dynamic routing does not achieve the top rank on every benchmark. Specifically, it ranked first on the MAWPS, AddSub, and SingleEq datasets, second on GSM8K, and third on AQuA and MultiArith. This performance variation implies that different branches may have developed specialized capabilities for distinct scenarios during training. Although dynamic routing strategy fails to select the optimal inference trajectory in certain cases, it effectively leverages the strengths of each branch across a wide range of scenarios, lead-

Table 3: Performance comparison between a 4-branch MoDr with dynamic routing and its four individual branches (Branch-($b$), $b \in \{1, 2, 3, 4\}$). Avg.Br-(1~4) denotes the average score of branches 1 to 4. The top score in each column is in **bold**, and the second-highest is underlined.

| Method | In Domain | | | Out of Domain | | | Average |
|---|---|---|---|---|---|---|---|
| | GSM8K | MAWPS | AQuA | MultiArith | AddSub | SingleEq | |
| MoDr (Ours) | 49.89 | **80.67** | 33.07 | 91.17 | **79.24** | **81.30** | **69.22** |
| ↳ Branch-(1) | **50.19** | 75.63 | **35.43** | 91.67 | 71.39 | 75.98 | 66.72 |
| ↳ Branch-(2) | 48.07 | 79.37 | 34.65 | 89.17 | 75.19 | 80.71 | 67.86 |
| ↳ Branch-(3) | 47.84 | 80.25 | 28.35 | **92.50** | 77.97 | 80.71 | 67.94 |
| ↳ Branch-(4) | 49.66 | 74.37 | 26.77 | 86.83 | 70.13 | 78.15 | 64.32 |
| ↳ Avg.Br-(1~4) | 48.94 | 77.41 | 31.30 | 90.04 | 73.67 | 78.89 | 66.71 |

ing to superior overall performance compared to any individual branch or their simple average. This confirms the critical role of a multi-branch dynamic routing framework in developing effective depth-recurrent reasoning models.

**Impact of Load Balance.** An unbalanced branch load can lead to routing collapse (Shazeer et al., 2017), causing an imbalance in branch utilization and diminishing computational efficiency. To mitigate this, we employ a load balancing strategy governed by the update rate $\eta$ in Eq. 19, which controls the convergence rate of the branch biases $\{b_i\}_{i=1}^{N}$ to a suitable bias. To evaluate this strategy, we conducted an ablation study by comparing a model with the strategy enabled ($\eta$=0.001) against a baseline without it ($\eta$=0). We introduce a metric called balance entropy to quantify the degree of load balance as follows:

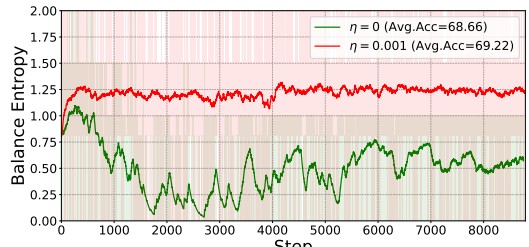

Figure 4: The impact of update rate on training load balance. Lower entropy makes it easier for the router to choose the same branch.

$$H_{\text{balance}} = - \sum_{br \in \text{Unique}(\Omega)} \frac{\text{Count}(br)}{|\Omega|} \log_2 \frac{\text{Count}(br)}{|\Omega|}, \qquad (20)$$

where $\Omega$ is the set of branches selected by the router within a batch, and $br$ is an element of this set. A higher entropy indicates a more balanced load distribution. As shown in Figure 4, the model with $\eta$=0 (without load balance) converges to a limited subset of branches, resulting in their disproportionate over-utilization. In contrast, the model with $\eta$=0.001 ensures a more even distribution of the training load across all branches, which ultimately leads to superior generalization performance.

## 4.4 SENSITIVITY ANALYSIS OF RECURRENT BRANCH NUMBERS

The number of recurrent branches is a critical hyperparameter, influencing both the search space size and computational resource allocation. To investigate its impact, we conducted a series of experiments where we varied the number of branches while keeping all other settings constant. To guarantee a fair comparison, the total number of training epochs was scaled proportionally with the number of branches, ensuring that each branch received an equivalent amount of computation. The results, as shown in Figure 5, reveal a clear trend: average performance across all benchmarks consistently improves as more branches are added. This positive correlation between branch count and performance validates the effectiveness of our dynamic multi-branch architecture. How-

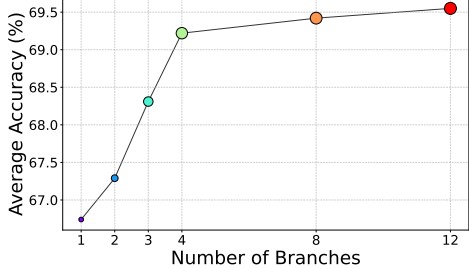

Figure 5: The changes of average performance under different recurrent branch numbers.

count and performance validates the effectiveness of our dynamic multi-branch architecture. How-

ever, we also observe that the performance gains begin to diminish beyond four branches, suggesting that this configuration offers the best trade-off between performance and computational cost.

## 4.5 QUANTITATIVE ANALYSIS OF BRANCH ROUTING

To gain insights into the routing selection distribution across different tasks, we conducted a frequency analysis on six mathematical reasoning benchmarks, as shown in Figure 6. The results reveal two key observations. First, the routing distribution is highly sensitive to the specific dataset. On complex, multi-step reasoning tasks like MultiArith, Branch-3 is activated at a significantly higher frequency (46.72%) than on other tasks. In contrast, for the multiple-choice question set AQuA, Branch-4 emerges as the second most utilized branch (33.16%). This indicates that the router dynamically allocates computational resources in response to the unique demands of each task. Second, the branches exhibit a clear division of labor. Branch-2 serves as a "generalist," consistently handling the majority of computations across all benchmarks. The remaining branches function as "specialists." For instance, Branch-3 is predominantly activated for complex arithmetic tasks (MAWPS, AddSub), whereas Branch-4 is more engaged in tasks requiring distinct reasoning patterns, such as those in AQuA. This non-uniform distribution provides strong quantitative evidence that the branches have learned distinct, non-redundant functions.

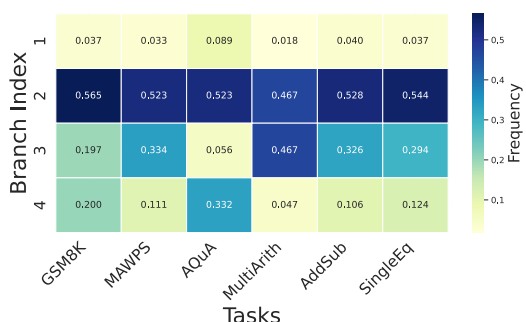

Figure 6: The changes of average performance under different recurrent branch numbers.

## 4.6 CASE STUDY

We further check more cases of the reasoning processes of MoDr. We find that at different reasoning steps during inference, MoDr dynamically invokes diverse branches to predict the next token based on the available context state, which is consistent with our hypothesis. Due to space limitations, more details are shown in Appendix A.3.

## 5 CONCLUSION

In this paper, we introduce the **M**ixture-**o**f-**D**epth-**R**ecurrent (MoDr) Transformer, a novel dynamic routing framework that advances the depth-recurrent Huginn model. The vanilla Huginn model's reasoning flexibility is constrained by its reliance on a single, chain-like propagation mechanism within the rumination recurrent module. MoDr addresses this limitation by incorporating multiple LoRA branches and employing a hard-gate router to dynamically select the most appropriate branch for next-token prediction. Extensive experiments across a diverse set of mathematical and commonsense reasoning benchmarks demonstrate that MoDr can significantly improves upon the performance of the existing Huginn model while incurring negligible computational overhead.

## 5.1 LIMITATIONS & FUTURE WORK

MoDr offers a dynamic multi-branch framework for the depth-recurrent Huginn model (Geiping et al., 2025), designed to enhance the exploration capability and adaptivity of its rumination recurrent module (Loop) within the latent space. By leveraging LoRAs as distinct branches, our approach avoids significant computational overhead. However, for practical deployment, MoDr necessitates an efficient KV cache strategy, which remains a key challenge and a primary direction for future work. Inspired by (Geiping et al., 2025; Bae et al., 2025), we identify two promising solutions: (1) caching KV pairs from the most recent $k$ recurrent iterations under a fixed budget, or (2) caching the initial KV pairs and sharing them across all recurrent branches for subsequent reasoning steps.

ACKNOWLEDGMENTS

We thank anonymous reviewers for their valuable and insightful feedback. The computational resources for this research were supported by the DataCanvas Alaya NeW platform.

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

# A  APPENDIX

## A.1  RELATED WORK

### A.1.1  TEST-TIME REASONING

Recent advances have demonstrated that large language models emerge with remarkable reasoning capabilities. In particular, Chain-of-Thought (CoT) reasoning (Wei et al., 2022b; Kojima et al., 2022) with few-shot or even zero-shot examples further achieves superior performance. These prompting techniques explicitly elicit intermediate reasoning steps before the model generates its final answers. Furthermore, several studies have further unlocked the potential of CoT when integrated with reinforcement learning-based fine-tuning (Jaech et al., 2024; Guo et al., 2025), uncovering significant "aha-moments" in model behavior. Alternatively, a compelling counterpart is to prompt LLMs that perform reasoning in latent space without verbal narrative. A strategy of this line is to explore auto-regressive latent reasoning by progressively replacing CoT tokens with continuous representations through multi-stage training (Hao et al., 2024), or compressing CoT tokens into continuous space via self-distillation in a single stage (Shen et al., 2025). Another line of work focuses on loop-based architectures, which iteratively refine hidden states in a single forward pass by recurrently propagating information across layers (Dehghani et al., 2019; Chen et al., 2025; Geiping et al., 2025). In addition, to achieve test-time depth adaptation, recent studies have explored methods for dynamically allocating computation in latent space. These methods learn to adapt to inputs of varying complexity by adjusting recursive depths (Bae et al., 2025; Chen et al., 2025), creating shortcuts (Wang et al., 2025), or combining both approaches (Li et al., 2025). Orthogonal to these works, we focus on developing versatile recurrent branches that can handle diverse contextual inputs, thereby improving the adaptability of the reasoning trajectory.

### A.1.2  PARAMETER-EFFICIENT FINE-TUNING

Supervised fine-tuning of large language models is a common practice for adaptation to various specific downstream tasks. However, as model parameters scale up, the computational cost of full fine-tuning becomes a significant challenge. To mitigate this, parameter-efficient fine-tuning (PEFT) methods (Houlsby et al., 2019; Li & Liang, 2021; Hu et al., 2022) have emerged over time. Low-rank adaptation (LoRA) (Hu et al., 2022) enables plug-and-play adaptation of pretrained LLMs by freezing initial model weights and fine-tuning a small set of low-rank matrices. For instance, (Gou et al., 2023) proposed a mixture of cluster-conditional LoRA experts to activate task-specific adapters based on instruction clusters. (Li et al., 2024) enhanced model performance by utilizing independent attention-layer LoRA adapters. Compared to full fine-tuning, this paradigm offers a more flexible and efficient solution with constrained computational resources. As for our approach, we integrate LoRAs as diverse exploration branches instead of simply copying the backbone recurrent module, thereby expanding model capacity and avoiding additional overhead.

### A.1.3  MULTI-BRANCH NETWORKS AND DYNAMIC ROUTING

Given that diverse inputs could have different computational demands, it is intuitive to perform inference with dynamic pathways tailored to each sample. Recent studies, such as MoCLE (Gou et al., 2023) and MoNE (Jain et al., 2024), aim to enhance model performance by maintaining multiple sub-networks and a learnable routing mechanism. The paradigm of Mixture-of-Experts (MoE) not only provides a promising solution for managing computational costs but also results in remarkable performance improvements. In conventional *soft* MoE (Eigen et al., 2013; Ma et al., 2018), the weights predicted by the router are adopted to dynamically aggregate the representations of all branches ("experts"). In contrast, *hard* MoE models (Shen et al., 2023; Guo et al., 2025) dynamically activate the branches with the highest or top-k confidence. While effective, these methods often encounter load imbalance, potentially leading to routing collapse (Shazeer et al., 2017). Therefore, introducing auxiliary balancing strategies (Fedus et al., 2022; Wang et al., 2024) is essential during the training phase. Inspired by these works, MoDr combines a hard-gate routing mechanism and an auxiliary-loss-free load balancing strategy to adaptively select the most suitable recurrent branch for the next reasoning step, thereby improving robustness while retaining performance.

## A.2 EXPERIMENTS FOR COMMONSENSE REASONING

### A.2.1 EXPERIMENTAL SETUP

**Datasets.** We conduct our experiments on six commonsense reasoning tasks: (1) **PIQA** (Bisk et al., 2020), a dataset focused on physical commonsense question answering. (2) **HellaSwag** (Zellers et al., 2019), a commonsense natural language interface (NLI) dataset. (3) **WinoGrande** (Sakaguchi et al., 2021), a dataset for commonsense reasoning involving pronoun disambiguation and sentence completion. (4) **ARC-E** and (5) **ARC-C** (Clark et al., 2018), the Easy and Challenge sets of the ARC dataset, which contains genuine grade-school level, multiple-choice science questions. (6) **OBQA** (Mihaylov et al., 2018), a dataset requiring multi-step reasoning, the use of additional common knowledge, and rich text comprehension. The detailed statistics of the datasets are outlined in Table 4. To facilitate fine-tuning in the domain of commonsense reasoning, we adopt the Commonsense170K [2] dataset constructed by Hu et al. (2023), and conduct evaluations on the individual testing dataset for each task.

**Other Protocols.** For the evaluation metric, comparison methods, and settings, we maintain consistency with the experiments on mathematical reasoning tasks (See §4.1).

Table 4: Statistics of Commonsense Reasoning Datasets.

| Dataset | PIQA | HellaSwag | WinoGrande | ARC-E | ARC-C | OBQA |
|---------|------|-----------|------------|-------|-------|------|
| Answer Type | Option | Option | Option | Option | Option | Option |
| # Train Sample | 16.1K | 39.9K | 63.2K | 1.1K | 2.3K | 5.0K |
| # Test Sample | 1830 | 10042 | 1267 | 2376 | 1172 | 500 |

### A.2.2 RESULTS AND ANALYSIS

Figure 7 illustrates a comparison of our proposed MoDr against the baseline models (i.e., vanilla Huginn and Huginn-SFT) on six commonsense reasoning tasks. MoDr demonstrates consistent performance gains over its competitors across most commonsense tasks, with the exception of HellaSwag, where it underperforms Huginn-SFT by 0.83%. In terms of average accuracy, MoDr achieves improvements of +21.21% and +1.52% over vanilla Huginn and Huginn-SFT, respectively. Overall, these results further verify MoDr's effectiveness on commonsense reasoning tasks.

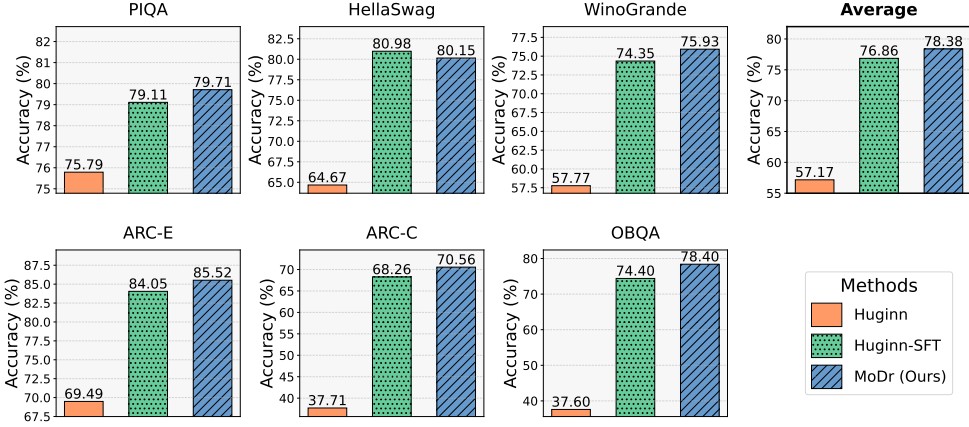

Figure 7: Performance comparison of MoDr with baseline methods on commonsense reasoning benchmarks. The Huginn results are reported by (Geiping et al., 2025). The "Average" represents the overall mean of the six commonsense datasets.

---

[2] https://huggingface.co/datasets/zwhe99/commonsense_170k

## A.3 CASE STUDY

We present three cases as follows.

> **Question:** James decides to run 3 sprints 3 times a week. He runs 60 meters each sprint. How many total meters does he run a week?
>
> ---
>
> **Answer:** To determine the total number of meters James runs in a week, we need to follow these steps:
>
> 1. Calculate the number of sprints James runs in a week.
> 2. Determine the total distance run in one sprint.
> 3. Multiply the number of sprints by the total distance run in one sprint to find the total distance run in a week.
>
> First, let's calculate the number of sprints James runs in a week. He runs 3 sprints 3 times a week. Therefore, the total number of sprints is:
> $$3 \times 3 = 9$$
>
> Next, we need to determine the total distance run in one sprint. Each sprint is 60 meters. So, the total distance run in one week is:
> $$9 \times 60 = 540$$
>
> Thus, the total number of meters James runs in a week is:
> $$\boxed{540}$$

Figure 8: Case study: The dynamic trajectories of text tokens across various reasoning branches in MoDr, color-coded as 1, 2, 3, and 4, for a test instance from **GSM8K**.

> **Question:** Lana was unboxing some of her old winter clothes. She found 3 boxes of clothing and inside each box there were 3 scarves and 4 mittens. How many pieces of winter clothing did Lana have total?
>
> ---
>
> **Answer:** To determine the total number of pieces of winter clothing Lana has, we need to follow these steps:
>
> 1. Calculate the number of scarves in the boxes.
> 2. Calculate the number of mittens in the boxes.
> 3. Add the number of scarves and mittens together.
>
> First, let's calculate the number of scarves in the boxes. Each box contains 3 scarves, and there are 3 boxes. Therefore, the total number of scarves is:
> $$3 \text{ boxes} \times 3 \text{ scarves per box} = 9 \text{ scarves}$$
>
> Next, let's calculate the number of mittens in the boxes. Each box contains 4 mittens, and there are 3 boxes. Therefore, the total number of mittens is:
> $$3 \text{ boxes} \times 4 \text{ mittens per box} = 12 \text{ mittens}$$
>
> Now, we add the number of scarves and mittens together to find the total number of pieces of winter clothing:
> $$9 \text{ scarves} + 12 \text{ mittens} = 21 \text{ pieces of winter clothing}$$
>
> Thus, the total number of pieces of winter clothing Lana has is:
> $$\boxed{21}$$

Figure 9: Case study: The dynamic trajectories of text tokens across various reasoning branches in MoDr, color-coded as 1, 2, 3, and 4, for a test instance from **MultiArith**.

From these cases, we have two interesting findings:

- During each inference process, not all branches participate throughout the entire lifecycle. This is shown in the first two cases.

- Branch 2 handles most of the upfront analysis and planning, while Branches 1, 3, and 4 focus on detailed reasoning and reaching conclusions. This is demonstrated across all cases.

This in turn, corroborates our hypothesis: distinct contextual inputs necessitate different computational demands, a dynamic mechanism for invoking specialized branches should be employed for next-token prediction.

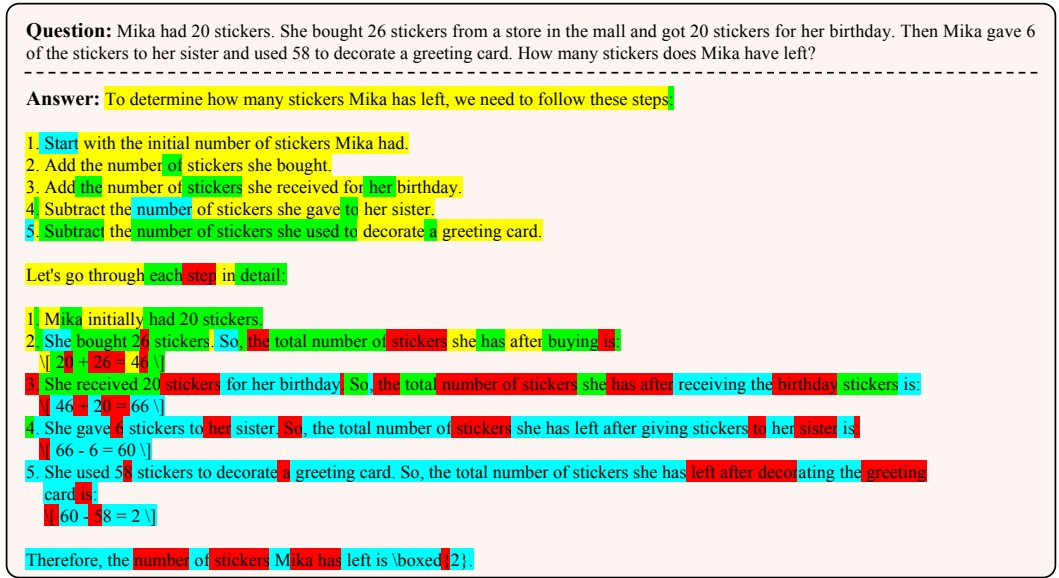

Figure 10: Case study: The dynamic trajectories of text tokens across various reasoning branches in MoDr, color-coded as **1**, **2**, **3**, and **4**, for a test instance from **SingleEq**.

## A.4 ADDITIONAL EXPERIMENTS FOR CODE GENERATION

### A.4.1 EXPERIMENTAL SETUP

To evaluate the model's code capabilities, we fine-tuned it on the CodeFeedback-Filtered-Instruction[3] (Zheng et al., 2024) dataset. After cleaning the data to remove invalid entries, we obtained a final training set of 79K samples. For evaluation, we used the HumanEval (Chen et al., 2021) and MBPP (Austin et al., 2021) test sets, adopting Pass@1 as the evaluation metric. The baseline methods and experimental settings were kept consistent with those used for the mathematical reasoning tasks (see Section 4.1).

### A.4.2 RESULTS AND ANALYSIS

Figure 11 presents the pass@1 scores on the HumanEval and MBPP benchmarks. As shown, our proposed MoDr achieves substantial and consistent improvements over all baselines in average performance, suggesting its potential for scalability. These results demonstrate that MoDr effectively extends the reasoning capabilities of the depth-recurrent Huginn model across a wide range of tasks, achieved by constructing a dynamic-routing, multi-branch mechanism within its recurrent rumination module.

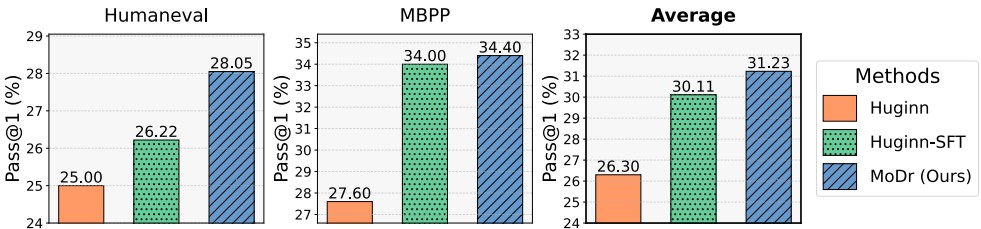

Figure 11: Performance (Pass@1) comparison of MoDr with baseline methods on code generation. The "Average" represents the overall mean of the HumanEval and MBPP datasets.

---

[3]https://huggingface.co/datasets/m-a-p/CodeFeedback-Filtered-Instruction

## A.5    Comparison to Standard Transformer Models of Similar Size

As depicted in Table 5, we compare our proposed MoDr with several open-weight standard Transformer models of similar scale. It is worth noting that some of the open-weight baselines are trained on more computational resources (10-15x). The open-weight models results are reported by (Geiping et al., 2025). These results further validate the advantages of our proposed multi-branch dynamic routing mechanism within the depth-recurrent transformer architecture.

Table 5: Performance comparison of MoDr with standard transformer models of similar size on commonsense reasoning benchmarks. The top score in each column is in **bold**.

| Model | Param | Tokens | PIQA | HellaSwag | WinoGrande | ARC-E | ARC-C | OBQA | Average |
|-------|-------|--------|------|-----------|------------|-------|-------|------|---------|
| Pythia-2.8B | 2.8B | 0.3T | 73.29 | 59.17 | 57.85 | 58.00 | 32.51 | 35.40 | 52.70 |
| OLMo-7B | 7B | 2.75T | **80.69** | 77.76 | 67.17 | 74.28 | 43.43 | 41.60 | 64.16 |
| Qwen2.5-3B | 3B | 18T | 78.18 | 55.02 | 67.80 | 77.23 | 44.54 | 29.40 | 58.70 |
| Llama3.2-3B | 3B | 9T | 76.77 | 55.20 | 69.22 | 74.58 | 42.58 | 31.00 | 58.30 |
| Llama-3.1-8B | 8B | 15T | 80.20 | 59.99 | 73.72 | 81.73 | 51.02 | 33.00 | 63.28 |
| Huginn-3.5B | 3.5B | 0.8T | 75.79 | 64.67 | 57.77 | 69.49 | 37.71 | 37.60 | 57.17 |
| MoDr (Ours) | 3.5B | 0.8T | 79.71 | **80.15** | **75.93** | **85.52** | **70.56** | **78.40** | **78.38** |

## A.6    Comparison to CoT, Majority Voting, and Full Fine-tuning

For a comprehensive performance evaluation, we also compared our proposed MoDr against several widely-used baselines, including few-shot Chain-of-Thought (CoT) prompting, Majority Voting (MV), and full fine-tuning (Full-FT). For the CoT and MV baselines, we adopted an eight-shot prompting configuration. As shown in Table 6, the results indicate that MoDr demonstrates significant performance advantages over all the baselines.

Furthermore, we made two key findings. First, compared to zero-shot prompting with the "reason step by step" instruction, the few-shot prompting does not improve the performance of the Huginn model. This may be attributed to the model's latent reasoning mechanism, which appears to respond better to implicit guidance. Second, we found that the performance of full fine-tuning remains suboptimal despite the convergence of the training loss curve. This suggests that LoRA fine-tuning is a more stable and effective fine-tuning method for the Huginn model.

Table 6: Performance of MoDr against baselines on mathematical reasoning benchmarks. We compare against eight-shot Chain-of-Thought (CoT), CoT with Majority Voting (CoT-MV), and Full Fine-Tuning (Full-FT). The best score in each column is **bold**.

| Method | In Domain | | | Out of Domain | | | Average |
|--------|-----------|------|------|------------|--------|----------|---------|
| | GSM8K | MAWPS | AQuA | MultiArith | AddSub | SingleEq | |
| CoT | 37.23 | 72.69 | 27.95 | 76.33 | 74.18 | 73.03 | 60.24 |
| CoT-MV | 37.76 | 79.41 | 29.53 | 81.17 | 79.49 | 74.02 | 63.56 |
| Full-FT | 6.67 | 50.00 | 6.69 | 78.83 | 65.32 | 47.83 | 42.56 |
| MoDr (Ours) | **49.89** | **80.67** | **33.07** | **91.17** | **79.24** | **81.30** | **69.22** |

## A.7    Expanded Results of Challenging Reasoning Benchmark

We conducted additional evaluation on the Super-GPQA (Du et al., 2025) benchmark, a composite dataset curated to be challenging across approximately 285 graduate disciplines. As shown in Table 7, our MoDr achieves the best performance and significantly outperforms the base Huginn model, which performs nearly at a random level. This demonstrates the effectiveness of our proposed MoDr even on this out-of-domain challenging benchmark. However, we also acknowledge that the absolute performance of both Huginn-SFT and MoDr on this highly challenging dataset is not yet satisfactory. We attribute this to two main factors: first, the relatively small scale of the base Huginn-3.5B model; second, the limited pre-training corpus of 0.8T tokens compared to the 10T+ tokens typically used for modern LLMs. These limitations likely constrain the performance of MoDr on

highly challenging reasoning tasks. Therefore, exploring the mixture-depth recurrent architecture on more advanced Recurrent Transformer models to achieve better generalization remains a promising direction for future research.

Table 7: Performance on Super-GPQA benchmark. The best score in each column is **bold**.

| Benchmark | Random | Huginn | Huginn-SFT | MoDr (Ours) |
|---|---|---|---|---|
| Super-GPQA | 10.59 | 10.63 | 14.07 | **14.40** |

## A.8 ANALYSIS OF TOP-K ROUTER

MoDr is flexible enough to support Top-K routing, in addition to its default Top-1 hard-gate router. To validate the impact of the number of routing branches (Top-K) on MoDr 's performance, we conducted an ablation study on the Top-1 and Top-2 routing configurations. The results, shown in Table 8, indicate that Top-2 routing improves performance on most datasets, suggesting that aggregating information from multiple experts is beneficial. However, we observed a significant performance drop on the AQuA dataset. This highlights a trade-off: while Top-K routing enhances performance across diverse tasks, it may introduce instability or sub-optimal routing on certain benchmarks.

Table 8: Impact of the number of routing branches (Top-K) on the performance of MoDr. The top score in each column is in **bold**.

| Method | Top-K | GSM8K | MAWPS | AQuA | MultiArith | AddSub | SingleEq | Average |
|---|---|---|---|---|---|---|---|---|
| MoDr | 1 | 49.89 | 80.67 | **33.07** | 91.17 | 79.24 | 81.30 | 69.22 |
| | 2 | **50.27** | **84.45** | 28.35 | **96.67** | **85.82** | **84.06** | **71.60** |

## A.9 PERFORMANCE COMPARISON WITH SAME TRAINING DURATION

To further validate the advantages of LoRA+MoE, we conducted additional experiments to evaluate the contribution of the proposed design. Specifically, we fine-tuned Huginn-SFT, MoDr (Top-1), and MoDr (Top-2) under the same training budget. The results, detailed in Table 9, demonstrate that MoDr with Top-1 and Top-2 routing outperforms the Huginn-SFT baseline by +0.25% and +3.03%, respectively. Notably, the Top-2 variant exhibits superior multi-branch dynamic reasoning capabilities. These findings confirm that MoDr achieves substantial improvements with the same training duration, reinforcing our central claim: the performance gains are attributed to the LoRA+MoE design rather than increased training time.

Table 9: Performance comparison with Huginn-SFT on mathematical reasoning benchmarks. The top score in each column is in **bold**.

| Method | Top-K | MAWPS | AQuA | MultiArith | AddSub | SingleEq | Average |
|---|---|---|---|---|---|---|---|
| Huginn-SFT | - | 80.25 | 31.50 | 93.17 | 77.97 | 81.30 | 72.84 |
| MoDr | 1 | 80.67 | **33.07** | 91.17 | 79.24 | 81.30 | 73.09 |
| | 2 | **84.45** | 28.35 | **96.67** | **85.82** | **84.06** | **75.87** |

## A.10 DETAILS FOR EXPERIMENTS

In Table 10, we report the computational efficiency and resource requirements of different methods. From this table, it is evident that our MoDr introduces only a modest number of additional trainable parameters compared to Huginn-SFT, resulting in a comparable FLOPs count. Importantly, its deployment memory and inference speed remain on par with the baseline methods. This demonstrates that our proposed dynamic routing and multi-branch architecture improves performance at no additional cost to deployment memory or inference speed.

Table 10: Details for Experiments.

| Metric | Huginn | Huginn-SFT | MoDr (Ours) |
|---|---|---|---|
| Trainable Parameters | 3.56B | 2.03M | 8.13M |
| TFLOPs | 35.00 | 35.55 | 35.60 |
| Training Memory | N/A | 54GB | 54GB |
| Deployment Memory | 13GB | 13GB | 13GB |
| Training Duration | N/A | 3h 58min | 16h 26min |
| Inference Speed | 15 token/s | 14 token/s | 13 token/s |

## A.11  EVALUATION DETAILS

We conducted evaluation experiments on all models using the following system prompts. In all experiments, the `temperature` of the models was set to 0.0001, and the `max_tokens` parameter was set to 1024. The number of steps for the recurrent module was set to 16. In particular, the AQuA dataset sometimes fails to follow the specified instructions for the final answer (e.g., placing a numerical value in a box rather than the corresponding option), we also manually double-checked the gold answers.

---

**Evaluation Prompt for Mathematical Reasoning Tasks**

**System**: Please reason step by step, and put your final answer within $\boxed{}$.
**User**: {Question}
**Assistant**:

---

**Evaluation Prompt for Commonsense Reasoning Tasks**

**System**: Below is an instruction that describes a task. Write a response that appropriately completes the request.
**User**:
`### Instruction`: {Instruction}

`### Response`:
**Assistant**:

---

**Evaluation Prompt for Code Generation Tasks**

**System**: You are an exceptionally intelligent coding assistant that consistently delivers accurate and reliable responses to user instructions.
**User**:
**// For Humaneval**
`### Instruction`: Read the following function signature and docstring, and fully implement the function described. Your response should only contain the code for this function.

Function signature: {Example}

`### Response`:

**// For MBPP**
`### Instruction`: {Test}

Example Test Cases: {Test_List}

`### Response`:
**Assistant**:

---

## A.12  FINE-TUNING EXAMPLES

The following are example questions and solutions from the mathematical reasoning, commonsense reasoning, and code generation datasets used for fine-tuning in our experiments.

### A.12.1 MATHEMATICAL REASONING TASKS

---

**GSM8K**

**System**: Please reason step by step, and put your final answer within $\boxed{}$.

**User**: Natalia sold clips to 48 of her friends in April, and then she sold half as many clips in May. How many clips did Natalia sell altogether in April and May?

**Assistant**: To determine the total number of clips Natalia sold in April and May, we start by finding out how many clips she sold in May. According to the problem, Natalia sold half as many clips in May as she did in April. Since she sold 48 clips in April, we calculate the number of clips sold in May as follows:

$$\text{Clips sold in May} = \frac{48}{2} = 24$$

Next, we add the number of clips sold in April to the number of clips sold in May to find the total number of clips sold over the two months:

$$\text{Total clips sold} = \text{Clips sold in April} + \text{Clips sold in May} = 48 + 24 = 72$$

Therefore, the total number of clips Natalia sold in April and May is $\boxed{72}$.

---

**AQuA**

**System**: Please reason step by step, and put your final answer within $\boxed{}$.

**User**: Carrie likes to buy t-shirts at the local clothing store. They cost $9.95 each. One day, she bought 20 t-shirts. How much money did she spend? The options are: A)$150 B)$248.75 C)$199 D)$171.6 E)$190

**Assistant**: To determine how much money Carrie spent on t-shirts, we need to multiply the cost of one t-shirt by the number of t-shirts she bought. The cost of one t-shirt is $9.95, and she bought 20 t-shirts. Let's perform the multiplication step-by-step:
1. Multiply the cost of one t-shirt by the number of t-shirts:

$$9.95 \times 20$$

2. To make the multiplication easier, we can break it down:

$$9.95 \times 20 = (10 - 0.05) \times 20$$

3. Use the distributive property to multiply:

$$(10 - 0.05) \times 20 = 10 \times 20 - 0.05 \times 20$$

4. Calculate each part separately:
$$10 \times 20 = 200$$
$$0.05 \times 20 = 1$$

5. Subtract the second result from the first:

$$200 - 1 = 199$$

Therefore, Carrie spent $\boxed{C}$ dollars.

---

**MAWPS**

**System**: Please reason step by step, and put your final answer within $\boxed{}$.

**User**: For the fifth grade play, the chairs have been put into 27 rows with 16 chairs in each row. How many chairs have been put out for the play ?

**Assistant**: To determine the total number of chairs put out for the play, we need to multiply the number of rows by the number of chairs in each row. Here are the steps:
1. Identify the number of rows: 27 2. Identify the number of chairs in each row: 16 3. Multiply the number of rows by the number of chairs in each row: $27 \times 16$
Now, let's perform the multiplication:

$$27 \times 16 = 27 \times (10 + 6) = (27 \times 10) + (27 \times 6)$$

---

First, calculate $27 \times 10$:
$$27 \times 10 = 270$$
Next, calculate $27 \times 6$:
$$27 \times 6 = 162$$
Now, add the two results together:
$$270 + 162 = 432$$
Therefore, the total number of chairs put out for the play is $\boxed{432}$.

### A.12.2 COMMONSENSE REASONING TASKS

**PIQA**

**System**: Below is an instruction that describes a task. Write a response that appropriately completes the request.
**User**:
### Instruction: Please choose the correct solution to the question: When boiling butter, when it's ready, you can
Solution1: Pour it onto a plate
Solution2: Pour it into a jar
Answer format: solution1/solution2
### Response:
**Assistant**: the correct answer is $\boxed{\text{solution2}}$.

**HellaSwag**

**System**: Below is an instruction that describes a task. Write a response that appropriately completes the request.
**User**:
### Instruction: Please choose the correct ending to complete the given sentence: Removing ice from car: Then, the man writes over the snow covering the window of a car, and a woman wearing winter clothes smiles. then
Ending1: , the man adds wax to the windshield and cuts it. Ending2: , a person board a ski lift, while two men supporting the head of the person wearing winter clothes snow as the we girls sled. Ending3: , the man puts on a christmas coat, knitted with netting. Ending4: , the man continues removing the snow on his car.
Answer format: ending1/ending2/ending3/ending4
### Response:
**Assistant**: the correct answer is $\boxed{\text{ending4}}$.

**WinoGrande**

**System**: Below is an instruction that describes a task. Write a response that appropriately completes the request.
**User**:
### Instruction: Please choose the correct answer to fill in the blank to complete the given sentence: Ian volunteered to eat Dennis's menudo after already having a bowl because despised eating intestine.
Option1: Ian Option2: Dennis Answer format: option1/option2
### Response:
**Assistant**: the correct answer is $\boxed{\text{option2}}$.

**ARC-E**

**System**: Below is an instruction that describes a task. Write a response that appropriately completes the request.
**User**:

### Instruction: Please choose the correct answer to the question: Which factor will most likely cause a person to develop a fever?
Answer1: a leg muscle relaxing after exercise Answer2: a bacterial population in the bloodstream Answer3: several viral particles on the skin Answer4: carbohydrates being digested in the stomach
Answer format: answer1/answer2/answer3/answer4
### Response:
**Assistant**: the correct answer is ⟨answer2⟩.

---

### ARC-C

**System**: Below is an instruction that describes a task. Write a response that appropriately completes the request.
**User**:
### Instruction: Please choose the correct answer to the question: George wants to warm his hands quickly by rubbing them. Which skin surface will produce the most heat?
Answer1: dry palms Answer2: wet palms Answer3: palms covered with oil Answer4: palms covered with lotion
Answer format: answer1/answer2/answer3/answer4
### Response:
**Assistant**: the correct answer is ⟨answer1⟩.

---

### OBQA

**System**: Below is an instruction that describes a task. Write a response that appropriately completes the request.
**User**:
### Instruction: Please choose the correct answer to the question: The sun is responsible for
Answer1: puppies learning new tricks Answer2: children growing up and getting old Answer3: flowers wilting in a vase Answer4: plants sprouting, blooming and wilting
Answer format: answer1/answer2/answer3/answer4
### Response:
**Assistant**: the correct answer is ⟨answer4⟩.

## A.12.3 CODE GENERATION TASKS

### CodeFeedback

**System**: You are an exceptionally intelligent coding assistant that consistently delivers accurate and reliable responses to user instructions.
**User**:
### Instruction: Write a recursive function that calculates the factorial of a given number in Python. The function should handle negative numbers as well. Additionally, the function should only accept non-negative integers as input and raise a ValueError for any other input. The function should also check if the given number is a prime number, and if so, return the factorial of the next prime number greater than the given number.
### Response:
**Assistant**: Here's a recursive function in Python that calculates the factorial of a given number, handles negative numbers, and raises a ValueError for any non-negative integers:

```python
def factorial(n):
    if not isinstance(n, int) or n < 0:
        raise ValueError("Input must be a non-negative integer")
    if n <= 1:
        return 1
    else:
        return n * factorial(n-1)
```

