# OpenReview forum: "MoDr: Mixture-of-Depth-Recurrent Transformers for Test-Time Reasoning"
_ICLR.cc/2026/Conference — ICLR 2026 Poster_

### Official Review · Reviewer_WUhc · 2025-10-25

**Soundness:** 2
**Presentation:** 3
**Contribution:** 2
**Rating:** 4
**Confidence:** 4

**Summary:**

In this work, authors propose the Mixture-of-Depth-Recurrent Transformer (MoDr), which replaces the single recurrent loop in the Huginn model with multiple LoRA-based branches. To switch between branches authors propose a hard-gated dynamic routing mechanism that adaptively selects the best branch for each reasoning step. This allows the model to explore diverse reasoning paths in latent space efficiently. A loss-free load balancing mechanism ensures that routing remains stable and evenly distributed across branches.
Authors test their model on several math and commonsense reasoning tasks and show improvements over vanilla model and LoRA finetuning.

**Strengths:**

1. The paper in well organized and easy to follow
2. Authors propose a novel architecture, combining MoE with dynamic routing, LoRA-based adapters, and recurrent nature of the Huginn model.
3. Authors explore an interesting paradigm for “dynamic latent exploration” during test-time reasoning, bridging the gap between latent reasoning and mixture-of-experts routing.
4. Authors show improvements on six mathematical and six commonsense reasoning datasets, covering both in-domain and out-of-domain settings.

**Weaknesses:**

1. While overall work is novel, all its components - Depth recurrence, MoE, and auxiliary-loss-free load balancing, - are assembled from existing paradigms rather than introducing a fundamentally new principle
2. The comparison set is limited mostly to Huginn and LoRA-Huginn models:
2a. For fair comparison, authors should specify additional compute budget, specifically number of additional parameters, total number of trainable parameters, spent FLOPS, and change in inference time, if any. Since MoDr introduces routing and multiple LoRA branches, empirical runtime and memory cost comparisons with Huginn are essential.
2b. Simple Chain-of-Thought prompting, majority voting, and full-SFT baseline are missing, while all being popular and cheap techniques widely used in literature and on practice.
3. The paper only reports Top-1 routing; Top-k routing or soft gating could reveal trade-offs between performance and stability.
4. The paper’s claim that routing improves reasoning exploration is plausible but largely untested. While the authors include a qualitative case study, no quantitative evidence shows that different branches develop distinct reasoning skills or that routing adapts to task complexity.

**Questions:**

See "Weaknesses" section

---

> ### Author Response · Authors · 2025-11-24
>
> We thank the reviewer very much for the constructive and in-depth suggestions and appreciation of our work. We provide responses to the reviewer’s specific questions and concerns below.
>
> 1. **Comment**: *“While overall work is novel, all its components - Depth recurrence, MoE, and auxiliary-loss-free load balancing, - are assembled from existing paradigms rather than introducing a fundamentally new principle.”*
>
>     **Response**: While all the components seem to be built upon existing paradigms, our primary contribution is both novel and significant. We would like to clarify this in two parts:
>
>     - **A Novel Observation of a Fundamental Limitation**: Our work begins with a critical insight: we identify that one inherent weakness of the Huginn architecture is its `Loop' module. The single, chain-like propagation mechanism within this module severely restricts the diversity and exploration of reasoning trajectories. This observation of "chain-like reasoning bottleneck" in recurrent models is a novel contribution in itself.
>     - **A Novel and Principled Solution**: Building on this observation, we are the first to propose and validate the use of dynamic branching routing as a solution. This is not a simple assembly. It is a targeted architectural innovation designed specifically to address the identified bottleneck. By replacing the chain with a dynamic routing mechanism, we fundamentally change how the model explores its latent space, enabling more complex and effective reasoning.
>
>     Therefore, our work’s novelty stems from this principled observation and a targeted integration that unlocks new capabilities for recurrent models.
>
>
> 2. **Comment**: *“The comparison set is limited mostly to Huginn and LoRA-Huginn models: 2a. For fair comparison, authors should specify additional compute budget, specifically number of additional parameters, total number of trainable parameters, spent FLOPS, and change in inference time, if any. Since MoDr introduces routing and multiple LoRA branches, empirical runtime and memory cost comparisons with Huginn are essential. 2b. Simple Chain-of-Thought prompting, majority voting, and full-SFT baseline are missing, while all being popular and cheap techniques widely used in literature and on practice.”*
>
>     **Response**: Thank you for your suggestion. We report the compute budget in the following table. It can be seen that our MoDr introduces a modest number of additional trainable parameters compared to Huginn-SFT, resulting in a comparable FLOPs count. Crucially, the deployment memory and inference speed remain on par with the baseline methods. This demonstrates that our proposed dynamic routing and multi-branch architecture improves performance at no additional cost to deployment memory or inference speed. We have included this in the revised paper (See Appendix A.10).
>
>     | Metric              | Huginn     | Huginn-SFT | MoDr (Ours) |
>     |---------------------|------------|------------|-------------|
>     | Trainable Parameters| 3.56B      | 2.03M      | 8.13M       |
>     | TFLOPs              | 35.00      | 35.55      | 35.60       |
>     | Training Memory     | N/A        | 54GB       | 54GB        |
>     | Deployment Memory   | 13GB       | 13GB       | 13GB        |
>     | Training Duration   | N/A        | 3h 58min   | 16h 26min   |
>     | Inference Speed     | 15 token/s | 14 token/s | 13 token/s  |
>
>     Following your suggestion, we report the performance of 8-shot Chain-of-Thought (CoT), majority voting (CoT-MV) and Full-FT baselines on mathematical reasoning benchmarks, as shown below. The results show that our MoDr method still achieves superior performance, validating the effectiveness of our proposed architecture. We have updated these results in the revised paper (See Appendix A.6).
>
>     |Method | GSM8K | MAWPS | AQuA | MultiArith | AddSub | SingleEq |Average |
>     |---|---|---|---|---|---|---|---|
>     | CoT | 37.23 | 72.69 | 27.95 | 76.33 | 74.18 | 73.03 | 60.24 |
>     | CoT-MV | 37.76 | 79.41 | 29.53 | 81.17 | 79.49 | 74.02 | 63.56 |
>     | Full-FT | 6.67 | 50.00 | 6.69 | 78.83 | 65.32 | 47.83 | 42.56 |
>     | MoDr (Ours) | **49.89** | **80.67** | **33.07** | **91.17** | **79.24** | **81.30** | **69.22** |

---

> > ### Author Response · Authors · 2025-11-24
> >
> > 3. **Comment**: *“The paper only reports Top-1 routing; Top-k routing or soft gating could reveal trade-offs between performance and stability.”*
> >
> >     **Response**: We sincerely thank the reviewer for raising this this excellent point, which prompted us to perform a deeper analysis of our routing mechanism. We have conducted an ablation study on Top-1 and Top-2 routing, and the findings are quite revealing. We report the results in the following table.
> >
> >     | Method | Top-K | GSM8K | MAWPS | AQuA | MultiArith | AddSub | SingleEq | Average |
> >     |--------|-------|-------|-------|------|-------------|--------|----------|---------|
> >     | MoDr | 1 | 49.89 | 80.67 | **33.07** | 91.17 | 79.24 | 81.30 | 69.22 |
> >     | MoDr | 2 | **50.27** | **84.45** | 28.35 | **96.67** | **85.82** | **84.06** | **71.60** |
> >
> >     It can be seen that Top-2 routing improves performance on most datasets, suggesting that aggregating information from multiple experts is beneficial. However, we observed a significant performance drop on the AQuA dataset. This highlights a trade-off: while Top-K routing enhances performance across diverse tasks, it may introduce instability or sub-optimal routing on certain benchmarks. We will include this analysis in the revised manuscript (See Appendix A.8).

---

> > > ### Author Response · Authors · 2025-11-24
> > >
> > > 4. **Comment**: *“The paper’s claim that routing improves reasoning exploration is plausible but largely untested. While the authors include a qualitative case study, no quantitative evidence shows that different branches develop distinct reasoning skills or that routing adapts to task complexity.”*
> > >
> > >     **Response**: We thank the reviewer for this insightful comment. We would like to address the reviewer's concerns by elaborating on our position, acknowledging the challenges, and the further efforts we have made for quantitative evidence.
> > >
> > >     1. The reviewer correctly pointed out a fundamental challenge in the field of Mixture-of-Experts models, the inherent "black box" nature. The routing mechanism, often driven by high-dimensional and opaque gating functions, makes it notoriously difficult to definitively prove why a specific token is assigned to a particular expert. This interpretability issue is a known and active area of research [1-3]. In this paper, we present a qualitative case study in its current form as a preliminary and necessary step to clarify the "possibility" of this specialized model. From these cases, we have two interesting findings:
> > >
> > >         - During each inference process, not all branches participate throughout the entire lifecycle.
> > >         - Different branches differentiate to take on distinct functions. For example, some branches are better suited for the bulk of the initial analysis and planning, while others excel at detailed reasoning and drawing conclusions.
> > >
> > >         We believe those demonstrations are valuable as it provides the first piece of evidence and a concrete hypothesis for the community to investigate further.
> > >
> > >     2. To quantitatively validate our claim, we conducted a routing frequency analysis across all six mathematical reasoning benchmarks, with results shown in the table below. We can observe that:
> > >
> > >         | Model    | GSM8K  | MAWPS  | AQuA   | MultiArith | AddSub | SingleEq |
> > >         |----------|--------|--------|--------|------------|--------|----------|
> > >         | Branch-1 | 0.037 | 0.033 | 0.089 | 0.018     | 0.040 | 0.037   |
> > >         | Branch-2 | 0.565  | 0.523 | 0.523 | 0.467     | 0.528 | 0.544   |
> > >         | Branch-3 | 0.197 | 0.334 | 0.056 | 0.467     | 0.326  | 0.294   |
> > >         | Branch-4 | 0.200 | 0.111  | 0.332 | 0.047     | 0.106 | 0.124   |
> > >
> > >         - The routing distribution is highly sensitive to the specific dataset. On complex, multi-step reasoning tasks like MultiArith, Branch-3 is activated at a significantly higher frequency (46.72\%) than on other tasks. In contrast, for the multiple-choice question set AQuA, Branch-4 emerges as the second most utilized branch (33.16\%). This indicates that the router dynamically allocates computational resources in response to the unique demands of each task.
> > >         - The branches exhibit a clear division of labor. Branch-2 serves as a “generalist,” consistently handling the majority of computations across all benchmarks. The remaining branches function as “specialists.” For instance, Branch-3 is predominantly activated for complex arithmetic tasks (MAWPS, AddSub), whereas Branch-4 is more engaged in tasks requiring distinct reasoning patterns, such as those in AQuA. This non-uniform distribution provides strong quantitative evidence that the branches have learned distinct, non-redundant functions.
> > >
> > >         We have added this analysis to the revised manuscript to support our claims about improved reasoning exploration through adaptive routing (See Appendix A.9).
> > >
> > >         [1] Yang et al. "Mixture of Experts Made Intrinsically Interpretable." ICML 2025.
> > >
> > >         [2] Lo et al. "A Closer Look into Mixture-of-Experts in Large Language Models." NAACL 2025.
> > >
> > >         [3] Tang et al. "Unveiling Hidden Collaboration within Mixture-of-Experts in Large Language Models."  arXiv preprint arXiv:2504.12359 (2025).

---

### Official Review · Reviewer_kgDC · 2025-10-31

**Soundness:** 4
**Presentation:** 4
**Contribution:** 4
**Rating:** 8
**Confidence:** 3

**Summary:**

This paper proposes Mixture-of-Depth-Recurrent (MoDr) Transformer, a novel extension of the depth-recurrent Huginn model. The key idea is to enhance the adaptability and exploration capability of latent-space reasoning by transforming Huginn’s single recurrent reasoning path into a multi-branch, dynamically routed system. Specifically, the authors:

Introduce multiple LoRA-based recurrent branches that share the Huginn loop weights, providing lightweight diversity in reasoning behavior.

Employ a hard-gate dynamic router to select the most suitable branch per token based on hidden-state context.

Design an auxiliary-loss-free load balancing strategy to prevent routing collapse during training.

**Strengths:**

Novel extension of depth-recurrent architectures — The work builds directly on Huginn and introduces a meaningful architectural innovation: turning the single-chain recurrent loop into a mixture-of-recurrent-paths framework.

Dynamic routing in latent reasoning — The proposed hard-gate router for token-level branch selection is conceptually elegant and empirically validated, pushing the field toward adaptive latent reasoning rather than verbal CoT.

Parameter-efficient design — The use of LoRA branches preserves the efficiency of the original model while expanding reasoning capacity.

Comprehensive experiments — Results on both mathematical and commonsense reasoning tasks are convincing, and ablations (router removal, branch count, load balancing) are thorough and insightful.

Strong empirical gains with negligible overhead, making the approach practical for future LLM test-time reasoning research.

**Weaknesses:**

I am not very familiar with the previous work (i.e., Huginn), but I find this paper to be quite intuitive and well-motivated. The Huginn model repeatedly reuses a single core module for latent reasoning, which inherently limits the model’s exploration depth. In contrast, this paper introduces a gated routing mechanism combined with LoRA adapters to merge multiple parallel recurrent branches, thereby enhancing the reasoning capability of LLMs. Overall, this is an elegant and valuable piece of work.

However, I still have some critical questions:

The paper only reports results on Qwen/Qwen2.5-Math-7B-Instruct, which is a relatively small model. Have the authors attempted to test larger LLMs (e.g., Qwen3, Qwen2.5-72B)? Since the proposed approach seems computationally lightweight, restricting the experiments to a single small model weakens the empirical support for the paper’s claims.

The evaluation benchmark is not sufficiently comprehensive. Some challenging reasoning benchmarks, especially in mathematics and science (e.g., AIME, GPQA, Super-GPQA), are missing, which limits the generality of the results.

**Questions:**

None

---

> ### Author Response · Authors · 2025-11-24
>
> We highly appreciate your time and effort in reviewing this paper, and also thank you a lot for your constructive and encouraging comments on our work. We answer your questions below.
>
> 1. **Comment**: *“The paper only reports results on Qwen/Qwen2.5-Math-7B-Instruct, which is a relatively small model. Have the authors attempted to test larger LLMs (e.g., Qwen3, Qwen2.5-72B)? Since the proposed approach seems computationally lightweight, restricting the experiments to a single small model weakens the empirical support for the paper’s claims.”*
>
>     **Response**: We thank the reviewer for this insightful comment. We agree that scaling laws are crucial in LLM research, and we would like to provide a two-part clarification.
>
>     - **Experimental Setup**: We may need to clarify that our MoDr architecture is not applied to Qwen2 5-Math-7B-Instruct. Our method is implemented on a 3.5B Recurrent Transformer (Huginn). The Qwen model served a single, specific purpose: to generate high-quality chain-of-thought reasoning answers for our training data. It was not subject of our modification or evaluation.
>
>     - **Rationale for Data Generator Choice**: The selection of Qwen2.5-Math-7B-Instruct as the data generator was a deliberate decision based on performance-per-cost. The official Qwen-math technical report [1] shows its performance is very close to its 72B counterpart (e.g., a negligible 0.7-point gap on GSM8K: 95.2 vs. 95.9). This made the 7B model a highly cost-effective choice, allowing us to generate a large volume of high-quality data which we then rigorously filtered.
>
>     Regarding the scale of our base model (Huginn), we acknowledge it is currently 3.5B. This is a novel work, and at the time of our experiments, a 3.5B version was the only pretrained Huginn model public available. We agree that scaling up is a critical direction and plan to explore MoDr on larger models in our future research.
>
>     [1] Yang et al. "Qwen2. 5-math technical report: Toward mathematical expert model via self-improvement." arXiv preprint arXiv:2409.12122 (2024).
>
> 2. **Comment**: *“The evaluation benchmark is not sufficiently comprehensive. Some challenging reasoning benchmarks, especially in mathematics and science (e.g., AIME, GPQA, Super-GPQA), are missing, which limits the generality of the results.”*
>
>     **Response**: Thank you for your suggestion. For a quick response to your concern, we have conducted additional evaluation on the Super-GPQA benchmark, and report the results in the following table. We have included this in the revised paper (See Appendix A.7).
>
>     | Benchmark | Random | Huginn | Huginn-SFT | MoDr (Ours) |
>     |-----------|--------|--------|-------------|-----------------|
>     | Super-GPQA | 10.59 | 10.63 | 14.07 | **14.40** |
>
>     As shown in this results, our MoDr achieves the best performance and significantly outperforms the base Huginn model, which performs nearly at a random level. This demonstrates the effectiveness of our proposed MoDr even on this out-of-domain challenging benchmark.
>
>     However, we also acknowledge that the absolute performance of both Huginn-SFT and MoDr on this highly challenging dataset is not yet satisfactory. We attribute this to two main factors: first, the relatively small scale of the base Huginn-3.5B model; second, the limited pre-training corpus of 0.8T tokens compared to the 10T+ tokens typically used for modern LLMs. These limitations likely constrain the performance of MoDr on highly challenging reasoning tasks.
>
>     Therefore, exploring the mixture-depth recurrent architecture on more advanced Recurrent Transformer models to achieve better generalization remains a promising direction for future research.

---

### Official Review · Reviewer_o7Ko · 2025-11-01

**Soundness:** 3
**Presentation:** 3
**Contribution:** 3
**Rating:** 6
**Confidence:** 3

**Summary:**

The paper proposes MoDr, a Mixture-of-Depth-Recurrent Transformer that improves test-time reasoning by replacing Huginn's single recurrent block with multiple LoRA-based branches and a hard-gate router.

**Strengths:**

1. The proposed idea is both novel and compelling. The authors introduce an innovative multi-branch dynamic routing mechanism for recurrent transformers, leveraging LoRA adapters to create lightweight and diverse reasoning branches, all while sharing the backbone weights for efficiency.

2. The method demonstrates strong performance, outperforming baseline models not only on in-domain tasks but also showing robust generalization to out-of-domain benchmarks.

3. The paper presents a comprehensive analysis, including detailed ablation studies, router effectiveness evaluations, branch specialization insights, and load balancing assessments. These provide robust empirical evidence supporting the model’s effectiveness, adaptability, and practical value.

**Weaknesses:**

1. It would be helpful to include more details about the trained model, such as training duration, detailed parameter count, etc.

2. A comparison with a standard Transformer model of similar size would strengthen the evaluation and better highlight the effectiveness of the proposed approach.

3. Including memory usage and FLOPs would provide a clearer picture of the model’s efficiency and practical deployment cost.

**Questions:**

NA

---

> ### Author Response · Authors · 2025-11-24
>
> We thank the reviewer very much for the insightful comments. We appreciate this opportunity to address your concerns.
>
> 1. **Comment**: *“1. It would be helpful to include more details about the trained model, such as training duration, detailed parameter count, etc. 2. Including memory usage and FLOPs would provide a clearer picture of the model’s efficiency and practical deployment cost.”*
>
>     **Response**: Thank you for the valuable suggestion. We have now added a comprehensive table to the revised paper that includes the requested training details (See Appendix A.10). The key information is as follows:
>
>     | Metric              | Huginn     | Huginn-SFT | MoDr (Ours) |
>     |---------------------|------------|------------|-------------|
>     | Trainable Parameters| 3.56B      | 2.03M      | 8.13M       |
>     | TFLOPs              | 35.00      | 35.55      | 35.60       |
>     | Training Memory     | N/A        | 54GB       | 54GB        |
>     | Deployment Memory   | 13GB       | 13GB       | 13GB        |
>     | Training Duration   | N/A        | 3h 58min   | 16h 26min   |
>     | Inference Speed     | 15 token/s | 14 token/s | 13 token/s  |
>
>     From this table, it is evident that our MoDr introduces only a modest number of additional trainable parameters compared to Huginn-SFT, resulting in a comparable FLOPs count. Importantly, its deployment memory and inference speed remain on par with the baseline methods. This demonstrates that our proposed dynamic routing and multi-branch architecture improves performance at no additional cost to deployment memory or inference speed.
>
> 2. **Comment**: *“A comparison with a standard Transformer model of similar size would strengthen the evaluation and better highlight the effectiveness of the proposed approach.”*
>
>     **Response**: We thank the reviewer for this valuable suggestion. To strengthen our evaluation, we have included standard Transformer models of a similar scale as new baselines. It is worth noting that some of the open-weight baselines are trained on more computational resources (10-15x)[1]. The following table presents the experimental results. These results further validate the advantages of our proposed multi-branch dynamic routing mechanism within the depth-recurrent transformer architecture. These additions have been integrated into the revised paper (See Appendix A.5).
>
>     | Model | Param | Tokens | PIQA | HellaSwag | WinoGrande | ARC-E | ARC-C | OBQA | Average |
>     |-------|-------|--------|------|-----------|-------------|-------|-------|-------|---------|
>     | Pythia-2.8B | 2.8B | 0.3T | 73.29 | 59.17 | 57.85 | 58.00 | 32.51 | 35.40 | 52.70 |
>     | OLMo-7B | 7B | 2.75T | **80.69** | 77.76 | 67.17 | 74.28 | 43.43 | 41.60 | 64.16 |
>     | Qwen2.5-3B | 3B | 18T | 78.18 | 55.02 | 67.80 | 77.23 | 44.54 | 29.40 | 58.70 |
>     | Llama3.2-3B | 3B | 9T | 76.77 | 55.20 | 69.22 | 74.58 | 42.58 | 31.00 | 58.30 |
>     | Llama-3.1-8B | 8B | 15T | 80.20 | 59.99 | 73.72 | 81.73 | 51.02 | 33.00 | 63.28 |
>     | Huginn-3.5B | 3.5B | 0.8T | 75.79 | 64.67 | 57.77 | 69.49 | 37.71 | 37.60 | 57.17 |
>     | MoDr (Ours) | 3.5B | 0.8T | 79.71 | **80.15** | **75.93** | **85.52** | **70.56** | **78.40** | **78.38** |
>
>     [1] Geiping et al., "Scaling up Test-Time Compute with Latent Reasoning: A Recurrent Depth Approach." ICML Workshop, 2025.

---

### Official Review · Reviewer_eRKJ · 2025-11-03

**Soundness:** 2
**Presentation:** 2
**Contribution:** 3
**Rating:** 2
**Confidence:** 3

**Summary:**

This paper studies a modification to the recurrent-depth model Huginn aimed at improving performance on reasoning tasks. Specifically, it introduces MoDr: a set of multiple LoRA parameter groups and a routing mechanism that determines which LoRA parameter set will be active for a given token. After supervised finetuning on reasoning traces, MoDr outperforms the baseline model and the model with a single LoRA parameter set on mathematical and commonsense reasoning tasks.

**Strengths:**

**Originality**  The submission shows that the multiple branches of a MoDr model specialize in different areas, and the routing mechanism appropriately sends data to the right specialist to obtain high performance (Table 3). This analysis of MoDr's LoRA+routing approach supports this general direction's potential for impact. Moreover, the performance benefits on reasoning tasks are notable.

**Quality**  The baselines facilitate testing of the proposed approach’s components (e.g., the contribution of the router). The method's generalization to OOD tasks is well established.

**Clarity**  The paper is mostly well written and clearly communicates key points.

**Significance**  The overall idea to equip latent reasoning with different modes that a router can send tokens through is very interesting. As the authors point out, the potential for more efficient reasoning models makes this area of study significant.

**Weaknesses:**

I would be happy to raise my score if the following can be addressed:
- There might be a weakness regarding the baselines' training durations. Please see "Questions" for details.

- Important related work is not discussed in the main text and appears to be missing in the appendix. Clear discussion of prior work that uses MoE/routing + LoRA would help clarify the methodological novelty of the submission. Applying such an approach to Huginn-like models seems new, and the submission should clarify other novel aspects through such discussion.

A minor weakness is that some points are made multiple times (e.g. chain-like reasoning and loss-free balancing are mentioned >5 times each in the first few pages), and more efficient writing could make space for clearer discussion of related work in the main text. Currently, related work is primarily discussed in the appendix.

**Questions:**

- Please find and discuss related works to address the contextualization weakness above. Some examples might include StructMoE or Lottery Ticket Adaptation.

- Figure 1: Could this be modified to illustrate the proposed approach more clearly? In contrast, Figure 2 is very helpful and clear.

- Line 246: The “relay race” analogy is unclear. Could you please clarify what is meant? Modifying Figure 1 could help address this point, too.

- How is the introduced load balancing different from the approach proposed in Wang et al. (2024)?

- Line 316: If the number of training epochs is greater for models with recurrent branches than it is for baseline models, should you have an ablation study that trains baseline models for more epochs? Relatedly, Figure 5 shows that performance improves with the number of branches, but the number of epochs is also changing, so it's unclear how much of the benefit is from branch count (as opposed to epoch count).

---

> ### Author Response · Authors · 2025-11-24
>
> We highly appreciate your time and effort in reviewing this paper, as well as your encouraging and constructive comments on our work. Below, we address to your key concerns point by point.
>
> # Weaknesses Part
>
> 1. **Comment**: *“Important related work is not discussed in the main text and appears to be missing in the appendix. Clear discussion of prior work that uses MoE/routing + LoRA would help clarify the methodological novelty of the submission. Applying such an approach to Huginn-like models seems new, and the submission should clarify other novel aspects through such discussion.”*
>
>     **Response**: Thank you for raising these discussion points. Following your suggestion, we have added a new subsection (Sec 2.1) to the main text in the revision.
>
>     In this new section, we now explicitly discuss and differentiate our work from prior studies that combine MoE/routing + LoRA. This discussion clearly positions our work. Our approach is motivated by a critical insight into a fundamental limitation of the Huginn architecture: its Loop module employs a monolithic, chain-like propagation mechanism that severely constrains the diversity and exploration of reasoning trajectories. To address this flaw, we propose MoDr, a novel MoE+LoRA architecture that departs from the conventional *layer-wise* paradigm. Instead, MoDr integrates *block-wise* low-rank adapters directly into Huginn’s recurrent module—a novel recurrent-depth transformer—and couples them with a dynamic routing mechanism. This design fundamentally enhances the adaptability and richness of the model’s reasoning trajectories, establishing a new direction for adaptive reasoning in recurrent-depth models.
>
> 2. **Comment**: *“A minor weakness is that some points are made multiple times (e.g. chain-like reasoning and loss-free balancing are mentioned >5 times each in the first few pages), and more efficient writing could make space for clearer discussion of related work in the main text. Currently, related work is primarily discussed in the appendix.”*
>
>     **Response**: We appreciate the reviewer’s comment regarding the writing efficiency and the placement of related work. To improve writing efficiency, we have revised the manuscript to reduce the repetition of key phrases like “chain-like reasoning” and “loss-free balancing” (e.g., on Lines 17, 188, and 211). Regarding the placement of related work, and as stated in our previous response, we have restructured that section in the main text to better clarify the novelty of our contribution.

---

> > ### Author Response · Authors · 2025-11-24
> >
> > # Questions Part
> >
> > 1. **Comment**: *“Please find and discuss related works to address the contextualization weakness above. Some examples might include StructMoE or Lottery Ticket Adaptation.”*
> >
> >     **Response**: Thank you for your valuable suggestion and these related works. Following your advice, we have extended our related work discussion to include **StructMoE** and **Lottery Ticket Adaptation**. We now clarify how our dynamic, block-wise routing for Huginn’s recurrent module differs from these approaches, thereby strengthening the contextualization of our work.
> >
> > 2. **Comment**: *“Figure 1: Could this be modified to illustrate the proposed approach more clearly? In contrast, Figure 2 is very helpful and clear. Line 246: The “relay race” analogy is unclear. Could you please clarify what is meant? Modifying Figure 1 could help address this point, too.”*
> >
> >     **Response**: We appreciate the constructive comments on Figure 1 and the “relay race” analogy. We have addressed both points in the revised manuscript. Below we respond to each point separately.
> >
> >     - **On Figure 1**: Following your suggestion, we have revised the figure to enhance clarity and more effectively convey our proposed method. Specifically, we introduced a new line to clarify that in our proposed MoDr, each next token of Answer originates from a distinct recurrent branch.
> >     - **On the “relay race” analogy**: We thank the reviewer for highlighting the ambiguity. We have now clarified the text as follows: “During inference, in Huginn, each token is generated by a fixed recurrent module, whereas in MoDr, tokens are produced dynamically through a sequential relay of multiple recurrent branches.” This revised explanation is now supported by the clearer Figure 1, which illustrates this dynamic, multi-branch process.
> >
> > 3. **Comment**: *“How is the introduced load balancing different from the approach proposed in Wang et al. (2024)?”*
> >
> >     **Response**: The difference between our load balancing approach and that of Wang et al. (2024) can be summarized as follows:
> >
> >     | Aspect | Wang et al. (2024) | MoDr (Ours) |
> >     |---|---|---|
> >     | Problem Addressed | Layer-wise expert imbalance in MoE (FFN) layers. | Block-wise routing collapse among recurrent branches. |
> >     | Architectural Context | Standard Transformer with MoE layers. | Novel Mixture-of-Depth-Recurrent Transformer. |
> >     | Key Innovation | Applying a loss-free balancing strategy to MoE FFN layers. | **First** to apply this strategy to a recurrent-depth architecture to address a new type of collapse. |
> >
> >     Compared to Wang et al. (2024), we pioneer the application of a loss-free balancing strategy in a new architectural context to address the fundamentally different problem of block-wise routing collapse.
> >
> > 4. **Comment**: *“Line 316: If the number of training epochs is greater for models with recurrent branches than it is for baseline models, should you have an ablation study that trains baseline models for more epochs? Relatedly, Figure 5 shows that performance improves with the number of branches, but the number of epochs is also changing, so it's unclear how much of the benefit is from branch count (as opposed to epoch count).”*
> >
> >     **Response**: Thank you for pointing out this potentially confusing aspect. Our experimental design is based on the principle of ensuring equivalent training for each individual branch.
> >
> >     To achieve this, we set the number of epochs to match the number of recurrent branches to ensure that each branch in our MoDr model receives a comparable amount of training as the single branch in the baseline Huginn-SFT model. In MoDr, only one branch is activated per forward pass due to dynamic routing. This experimental setting ensures training equivalence in two ways: First, the computational cost per forward pass is nearly identical (35.55 TFLOPS for the baseline vs. 35.60 TFLOPS for MoDr, with the extra 0.05 TFLOPS attributed to the router). Second, matching epochs to branches guarantees that each branch undergoes the same total number of forward passes as the baseline model over the entire training process (noting that we freeze all non-branch parameters during fine-tuning). Consequently, the performance gains in Figure 5 stem from the increased capacity afforded by multiple branches, not from an unfair training advantage.

---

> ### Comment · Reviewer_eRKJ · 2025-11-24
>
> Thank you for improving the manuscript's clarity! I accordingly raised my score to 4 and would like to better understand the novelty of the submission (and the relevance of its baseline) before raising my score further.
>
> ## Novelty
> 1) Is it correct to say that the utilized load balancing technique previously existed but hadn't been tested with your block-wise LoRA approach in Huginn yet?
>
> 2) Relative to layer-wise LoRA, block-wise LoRA differs by applying LoRA only to the layers inside the recurrent blocks, or is there some other difference?
>
> ## Baseline
> I remain concerned that the baseline is not given equal training time. Could you please let me know if it's hard to run this test for some reason (e.g., limited compute)? To me, it's worth running this test to clarify that the benefits of the approach stem from LoRA+MoE rather than more training.

---

> > ### Author Response · Authors · 2025-11-26
> >
> > Thanks a lot for your reply. We respond to your new concerns below.
> >
> > 1. **Comment**: *Is it correct to say that the utilized load balancing technique previously existed but hadn't been tested with your block-wise LoRA approach in Huginn yet?”*
> >
> >    **Response**:  Thanks for this comment. The concept of a “loss-free” balancing strategy was indeed introduced by Wang et al. (2024). In this work, our main contribution lies in identifying that the original Huginn's rigid recurrent module constrains its reasoning capabilities and flexibility. To address this, we propose a novel dynamic multi-branch architecture for Huginn. A challenge in such multi-branch systems is imbalanced training caused by routing collapse. Inspired by Wang et al. and from a block-wise LoRA perspective, we therefore tailor the “loss-free” balancing strategy to our recurrent-depth architecture. This adaptation effectively addresses the new collapse problem of our design.
> >
> > 2. **Comment**: *Relative to layer-wise LoRA, block-wise LoRA differs by applying LoRA only to the layers inside the recurrent blocks, or is there some other difference?”*
> >
> >    **Response**: Thank you for this question. We appreciate this opportunity to clarify one key design choice in our work, stemming from a fundamentally distinct philosophy.
> >
> >    In layer-wise LoRA with MoE, the approach targets every single FFN layer in the Transformer. Within each of these layers, multiple low-rank adapters and a router form a sub-MoE module to distribute tokens among different LoRA experts. The focus is on enhancing the representational capacity of each individual layer.
> >
> >    In contrast, our block-wise LoRA does not target every layer. It aims to transform the monolithic single-path recurrent module into a mixture of reasoning paths. We use block-wise LoRA to create multiple, distinct latent reasoning branches, and a router dynamically selects which entire branch is best suited for processing the current sequence.
> >
> >    Therefore, the difference is not just in scope, but in the design paradigm: layer-wise LoRA aims for intra-layer expert specialization across the network, while our work aims for inter-path functional diversification to overcome the inflexibility of a single reasoning trajectory.
> >
> > 3. **Comment**: *“I remain concerned that the baseline is not given equal training time. Could you please let me know if it's hard to run this test for some reason (e.g., limited compute)? To me, it's worth running this test to clarify that the benefits of the approach stem from LoRA+MoE rather than more training.”*
> >
> >    **Response**: Thanks for this follow-up question. The core of the issue lies in how we define a “fair” comparison. There are two valid, yet conflicting, perspectives:
> >
> >    - **Per-Branch Training Alignment**: Our initial design was based on the principle that each individual branch in the MoDr model should receive a comparable amount training as the single branch in the baseline model. Because only one branch is activated per forward pass in our MoDr architecture, training for N epochs with N branches means that each individual branch receives exactly the same number of forward passes as the baseline model trained for N epochs. Training the baseline for N times more training than any single branch in our MoDr, which could be seen as an unfair advantage to the baseline.
> >    - **Total Training Duration Alignment**: This perspective argues that both models should be trained for the same total number of epochs. This is intuitive and understandable.
> >
> >    In response to this question, we conducted additional experiments to verify the contribution of the LoRA+MoE design. First, to establish a baseline, we fine-tuned Huginn-SFT given equal training epochs. Second, we recall that our MoDr model incorporates a key MoE feature: a top-k gate router that dynamically selects experts. To better demonstrate the advantages of LoRA+MoE, we further conducted an ablation study on top-2 routing with the same training budget. The results are reported in the following table.
> >
> >    | Method | Top-K | MAWPS | AQuA | MultiArith | AddSub | SingleEq | Average |
> >    |--|--|--|--|--|--|--|--|
> >    | Huginn-SFT | - | 80.25 | 31.50 | 93.17 | 77.97 | 81.30 | 72.84|
> >    | MoDr | 1 | 80.67 | 33.07 | 91.17 | 79.24 | 81.30 | 73.09 |
> >    | MoDr | 2 | 84.45 | 28.35 | 96.67 | 85.82 | 84.06 | 75.87 |
> >
> >    It can be seen that under the same number of training epochs, MoDr with Top-1 and Top-2 routing surpasses the Huginn-SFT baseline by +0.25\% and +3.03\% in average performance, respectively. Notably, MoDr with Top-2 routing more effectively demonstrates the advantages of multi-branch dynamic reasoning. Therefore, these extended experiments confirm that, with equal training duration, our MoDr achieves a significant performance advantage. This addresses the initial concern and reinforces our central claim: MoDr’s performance gain stems not from increased training but from the LoRA+MoE architectural design.

---

### Author Response · Authors · 2025-12-02

Dear Area Chair,

We sincerely appreciate your willingness to take on this additional workload and are grateful for your efforts. To facilitate your evaluation, we have prepared a brief, objective summary of the rebuttal interactions and the preliminary results achieved before the rebuttal phase was closed.
## Scores Status
Prior to the bug lack incident:
- The average score had risen from **5.0 to 5.5**.
- Reviewer (`eRKJ`) had been actively engaged with our rebuttal, updating their initial score（2 $\rightarrow$ **4** (25 Nov, 06:22)）and expressing an intention to raise it further (Stated: *I accordingly raised my score to 4 and would like to better understand the novelty of the submission (and the relevance of its baseline) before raising my score further.*). We had further addressed the reviewer’s concerns and were **awaiting their feedback before the leakage.**
- We have also addressed all the concerns from the other three reviewers (`o7Ko`: **6**,`kgDC`: **8**, and `WUhc`: **4**), **but discussion was concluded before they could respond.**
## Strengths Highlighted by Reviewers
- **Novelty**: `o7Ko` acknowledges the work is *“both novel and compelling,”* while `kgDC` describes it as *“an elegant and valuable piece of work.”*
- **Significance**: The overall idea is considered *“very interesting”* and *“significant”* for its potential to create more efficient reasoning models (`eRKJ`).  The work’s significance is further supported by `o7Ko`’s evidence of its *“practical value”* and `kgDC`’s note of its *“strong empirical gains with negligible overhead,”* highlighting its impact on future research.
- **Soundness**: Our work is praised for its *“comprehensive analysis”* (`o7Ko`) and *“convincing”* results with *“thorough and insightful”* ablations (`kgDC`). The baselines facilitate proper component testing and the model’s generalization to out-of-domain tasks is *“well established”* (`eRKJ`).
- **Clear Presentation**: The paper is *“mostly well written and clearly communicates key points,”* as noted by `eRKJ`, and is also described as *“well organized and easy to follow”* by `WUhc`.
## Summary of Key Revisions and New Results
1. **Enhanced Contextualization and Manuscript Clarity (Re: `eRKJ`)** To better position our contribution, we have introduced a new subsection (Sec. 2.1) that explicitly delineates our work from prior studies. Concurrently, we have enhanced manuscript clarity by reducing lexical redundancy and refining Figure 1 for improved visual communication.
1. **Comprehensive Computational Budget Analysis (Re: `o7Ko` and `WUhc`)** We now provide a comprehensive analysis of computational efficiency and resource requirements. A newly added table details key metrics—including trainable parameters, TFLOPs, memory footprint (training and deployment), training duration, and inference speed. This analysis empirically demonstrates that MoDr introduces a modest parameter overhead, yielding comparable FLOPs to the baseline while maintaining equivalent deployment memory and inference speed.
2. **Clarification of Novelty (Re: `eRKJ` and `WUhc`)** We have further clarified the novelty of our contributions:
    - For `eRKJ`, we specify that our load-balancing strategy uniquely addresses **block-wise routing collapse** within recurrent-depth architectures, which is fundamentally distinct from the **layer-wise expert** imbalance tackled in conventional MoE layers (e.g., Wang et al., 2024).
    - For `WUhc`, we emphasize our twofold contribution: **(1) a novel observation of a fundamental limitation** in existing methods and (2) **a principled and novel solution** to address it.
3. **Expanded and In-Depth Empirical Evaluation (Re: `eRKJ`, `o7Ko`, `kgDC` and `WUhc`)** To robustly validate our claims, we have significantly extended our experimental evaluation:
    - For `eRKJ`: An ablation study was conducted to isolate the performance gains attributable to the LoRA+MoE architectural design, decoupling them from the effects of increased training.
    - For `o7Ko`: Standard Transformer models of comparable scale were included as new, strong baselines.
    - For `kgDC`: Additional evaluation was performed on the challenging Super-GPQA benchmark.
    - For `WUhc`: We provide a multi-faceted analysis, including (1) comparisons against established baselines (few-shot CoT, Majority Voting, full fine-tuning), (2) an ablation study on Top-1 and Top-2 routing, and (3) a frequency analysis of routing selections across diverse reasoning tasks.

We have carefully considered each of the reviewers’ concerns and addressed them with reasonable justification and comprehensive experiments. All corresponding revisions have been incorporated into the revised manuscript. We believe these new experiments robustly validate our contributions.

We would like to once again thank you for your diligent efforts for navigating this challenging situation. We are available for further discussion at any time.

Best regards,

The Authors

---

### Meta-Review · Area_Chair_zTCS · 2026-01-07

**Summary:**

1) Reviewers raised questions about novelty and positioning relative to prior work combining LoRA with routing/MoE, and asked for clearer contextualization and a clearer articulation of what is new in the recurrent-depth setting.

2) There were concerns about fairness of baselines and training budget, specifically whether MoDr’s gains could be explained by more training rather than the LoRA+MoE routing design, and whether comparisons were missing common low-cost baselines.

3) Reviewers asked for stronger evidence around efficiency and mechanism, including explicit compute/memory/runtime reporting, comparisons to standard Transformer baselines, and more quantitative support for the claim that routing yields branch specialization / adaptive behavior (beyond qualitative examples).

**Reviewer Concerns:**

1) This was substantially addressed. The authors added a dedicated related-work/contextualization subsection (Sec. 2.2 but erroneously mentioned as 2.1 in the rebuttal) and explicitly distinguish their block-wise routing within a recurrent-depth module from layer-wise MoE/LoRA approaches. While the “novel observation + targeted solution” framing may still read somewhat interpretive, the rebuttal makes the novelty claim much clearer and better situated.

2) This was meaningfully addressed with new controls/ablations. The reviewer’s main worry (unequal training time) was directly tested via an added experiment training Huginn-SFT under equal epochs and reporting MoDr gains under matched budget, plus a Top-k routing ablation (Top-1 vs Top-2) under the same training regime. The authors also added common baselines requested by another reviewer (few-shot CoT, majority voting, full fine-tuning) and clarify the intended fairness criterion (per-branch vs total-epoch alignment). This does not eliminate all possible confounds, but it addresses the main concerns.


3) This was largely addressed with concrete reporting and added analyses. The rebuttal indicates a new table with parameter/FLOPs/memory/training time/inference speed, additional standard Transformer baselines of comparable scale, and quantitative routing analyses (routing frequency differences across datasets) plus branch/Top-k studies. One residual limitation is that the base Huginn model is currently only available at ~3.5B and trained on relatively modest pretraining tokens, so absolute performance and scaling claims remain constrained. Within that scope, however, the revised evaluation is much more complete and supports the paper’s central empirical claims.

**Reviewer Scores:**

the initial scores were 2,4,6,8

Reviewer eRKJ who gave the 2 alrady said they would increase to 4, and seeing how the rebuttal was going and can see that they would have even increased to a 6.

The 4, by Reviewer WUhc might have gone to a 6 but I doubt it.

---

### Decision · Program_Chairs · 2026-01-26

Accept (Poster)